# Skeletal dysplasia-causing TRPV4 mutations suppress the hypertrophic differentiation of human iPSC-derived chondrocytes

Amanda R Dicks[1,2,3], Grigory I Maksaev[4], Zainab Harissa[1,2,3], Alireza Savadipour[2,3,5], Ruhang Tang[2,3], Nancy Steward[2,3], Wolfgang Liedtke[6,7], Colin G Nichols[4], Chia-Lung Wu[8†], Farshid Guilak[2,3*†]

[1]Department of Biomedical Engineering, Washington University in St. Louis, St Louis, United States; [2]Department of Orthopedic Surgery, Washington University School of Medicine, St. Louis, St Louis, United States; [3]Shriners Hospitals for Children - St. Louis, St. Louis, United States; [4]Department of Cell Biology and Physiology, Washington University School of Medicine, St. Louis, St Louis, United States; [5]Department of Mechanical Engineering and Material Science, Washington University in St. Louis, St. Louis, United States; [6]Department of Neurology, Duke University School of Medicine, Durham, United States; [7]Department of Molecular Pathobiology - NYU College of Dentistry, New York, United States; [8]Department of Orthopaedics and Rehabilitation, Center for Musculoskeletal Research, University of Rochester, Rochester, United States

*For correspondence: guilak@wustl.edu

†co-senior author

**Abstract** Mutations in the TRPV4 ion channel can lead to a range of skeletal dysplasias. However, the mechanisms by which TRPV4 mutations lead to distinct disease severity remain unknown. Here, we use CRISPR-Cas9-edited human-induced pluripotent stem cells (hiPSCs) harboring either the mild V620I or lethal T89I mutations to elucidate the differential effects on channel function and chondrogenic differentiation. We found that hiPSC-derived chondrocytes with the V620I mutation exhibited increased basal currents through TRPV4. However, both mutations showed more rapid calcium signaling with a reduced overall magnitude in response to TRPV4 agonist GSK1016790A compared to wildtype (WT). There were no differences in overall cartilaginous matrix production, but the V620I mutation resulted in reduced mechanical properties of cartilage matrix later in chondrogenesis. mRNA sequencing revealed that both mutations up-regulated several anterior *HOX* genes and down-regulated antioxidant genes *CAT* and *GSTA1* throughout chondrogenesis. BMP4 treatment up-regulated several essential hypertrophic genes in WT chondrocytes; however, this hypertrophic maturation response was inhibited in mutant chondrocytes. These results indicate that the TRPV4 mutations alter BMP signaling in chondrocytes and prevent proper chondrocyte hypertrophy, as a potential mechanism for dysfunctional skeletal development. Our findings provide potential therapeutic targets for developing treatments for TRPV4-mediated skeletal dysplasias.

## Editor's evaluation

Analysis of different types of TRPV4 mutant hiPS cells (mild V620I vs severe T89I mutations) showed alterations in calcium channel function and chondrocyte differentiation. hiPSC-derived chondrocytes with the V620I mutation exhibited increased basal currents through TRPV4, while both mutations showed more rapid calcium signaling with a reduced overall magnitude in response to TRPV4

agonist GSK1016790A compared to wild-type cells. These findings provide potential therapeutic targets for developing treatments for TRPV4-mediated skeletal dysplasias.

## Introduction

Skeletal dysplasias comprise a heterogeneous group of over 450 bone and cartilage diseases with an overall birth incidence of 1 in 5000 (*Krakow and Rimoin, 2010*; *Nemec et al., 2012*; *Ngo et al., 2018*; *Orioli et al., 1986*; *Superti-Furga and Unger, 2007*). Mutations in transient receptor potential vanilloid 4 (TRPV4), a non-selective cation channel, can lead to varying degrees of skeletal dysplasia, including moderate autosomal-dominant brachyolmia and severe metatropic dysplasia (*Andreucci et al., 2011*; *Kang, 2012*). For example, a V620I substitution (exon 12, G858A) in TRPV4 is responsible for moderate brachyolmia, which exhibits short stature, scoliosis, and delayed development of deformed bones (*Kang et al., 2012*; *Rock et al., 2008*; *Kang, 2012*). These features, albeit more severe, are also present in metatropic dysplasia. Metatropic dysplasia can be caused by a TRPV4 T89I substitution (exon 2, C366T) and leads to joint contractures, disproportionate measurements, and, in severe cases, neonatal death due to small chest size and cardiopulmonary compromise (*Camacho et al., 2010*; *Kang et al., 2012*; *Kang, 2012*). Both V620I and T89I TRPV4 mutations are considered gain-of-function variants (*Leddy et al., 2014b*; *Loukin et al., 2011*). Given the essential role of TRPV4 during chondrogenesis (*Muramatsu et al., 2007*; *Willard et al., 2021*) and cartilage homeostasis (*O'Conor et al., 2014*), it is hypothesized that TRPV4 mutations may affect endochondral ossification during skeletal development.

Endochondral ossification is a process by which bone tissue is created from a cartilage template (*Breeland et al., 2021*; *Camacho et al., 2010*; *Krakow and Rimoin, 2010*; *Rimoin et al., 2007*). During this process, chondrocytes transition from maintaining the homeostasis of cartilage, regulated by transcription factor SRY-box containing gene 9 (*SOX9*) (*Breeland et al., 2021*; *Nishimura et al., 2012b*; *Prein and Beier, 2019*; *Sophia Fox et al., 2009*), to hypertrophy. Hypertrophy is driven by runt-related transcription factor 2 (*RUNX2*) and bone morphogenic protein (BMP) signaling (*Breeland et al., 2021*; *Nishimura et al., 2012b*; *Prein and Beier, 2019*) and leads to chondrocyte apoptosis or differentiation into osteoblasts to form bone (*Breeland et al., 2021*; *Nishimura et al., 2012b*; *Prein and Beier, 2019*). However, how TRPV4 and its signaling cascades regulate endochondral ossification remains to be determined.

The activation of TRPV4 increases *SOX9* expression (*Muramatsu et al., 2007*) and prevents chondrocyte hypertrophy and endochondral ossification (*Amano et al., 2009*; *Hattori et al., 2010*; *Lui et al., 2019*; *Nishimura et al., 2012a*; *Nishimura et al., 2012b*). One study found that overexpressing wildtype (WT) *Trpv4* in mouse embryos increased intracellular calcium ($Ca^{2+}$) concentration and delayed bone mineralization (*Weinstein et al., 2014*), a potential link between intracellular $Ca^{2+}$, such as with gain-of-function TRPV4 mutations, and delayed endochondral ossification. Our previous study also observed increased expression of follistatin (*FST*), a potent BMP inhibitor, and delayed hypertrophy in porcine chondrocytes overexpressing human V620I- and T89I-TRPV4 (*Leddy et al., 2014a*; *Leddy et al., 2014b*). While previous studies have greatly increased our knowledge of the influence of TRPV4 mutations on chondrogenesis and hypertrophy, most of them often involved animal models (*Leddy et al., 2014b*; *Weinstein et al., 2014*) or cells (*Camacho et al., 2010*; *Krakow and Rimoin, 2010*; *Leddy et al., 2014b*; *Loukin et al., 2011*; *Rock et al., 2008*) overexpressing mutant TRPV4. Therefore, these approaches may not completely recapitulate the effect of TRPV4 mutations on human chondrogenesis.

Human-induced pluripotent stem cells (hiPSCs), derived from adult somatic cells (*Takahashi et al., 2007*), offer a system for modeling human disease to study the effect of mutations throughout differentiation (*Adkar et al., 2017*; *Lee et al., 2021*). In fact, two studies have used patient-derived hiPSCs with TRPV4 mutations to study lethal and non-lethal metatropic dysplasia-causing variants I604M (*Saitta et al., 2014*) and L619F (*Nonaka et al., 2019*), respectively. However, patient samples are often challenging to procure due to the rarity of skeletal dysplasias. In this regard, CRISPR-Cas9 technology allows the creation of hiPSC lines harboring various mutations along with isogenic controls (i.e., WT).

The goal of this study was to elucidate the molecular mechanisms underlying how two TRPV4 gain-of-function mutations lead to strikingly distinct severities of skeletal dysplasias (i.e., moderate

brachyolmia vs. lethal metatropic dysplasia). To achieve this goal, we generated CRISPR-Cas9 gene-edited hiPSC lines bearing either the V620I or T89I TRPV4 mutation, and their isogenic WT control, to delineate the effects of TRPV4 mutations on chondrogenesis and hypertrophy using RNA sequencing and transcriptomic analysis. We further examined the effects of the mutations on channel function and matrix production and properties. We hypothesized the V620I and T89I TRPV4 mutations would enhance chondrogenesis with distinct degrees of altered hypertrophy. This study will improve our understanding of the role of TRPV4 in chondrocyte homeostasis and maturation and lay the foundation for treatment and prevention of TRPV4-mediated dysplasias.

## Results

### Mutant TRPV4 has altered response to chemical agonist GSK101

We first assessed TRPV4 channel function and alterations in $Ca^{2+}$ signaling due to the V620I and T89I mutations in day-28 hiPSC-derived chondrocytes using electrophysiology and fluorescence imaging. Using whole-cell patch clamping, we measured the basal membrane current of the hiPSC-derived chondrocytes from the mutated and WT lines. V620I-TRPV4 had the highest basal currents at both 70 and −70 mV (70/−70 mV pA/pF – WT: 18.52/5.93 vs. V602I: 77.79/55.33 vs. T89I: 40.97/50.13; *Figure 1A*). However, when TRPV4 was inhibited with GSK205 (*Kanju et al., 2016*), a TRPV4-specfic chemical antagonist, the three lines had similar, decreased currents (70/−70 mV – WT: 18.72/14.36 pA/pF vs. V620I: 13.55/9.15 pA/pF vs. T89I: 29.27/13.8 pA/pF; *Figure 1A*). To capture the specific current through TRPV4, we took the difference of the basal current (no GSK205) and the average TRPV4-inhibited current (with GSK205). TRPV4 inhibition caused a significant change in current in V620I at both 70 and −70 mV (70 mV – V620I: Δ64.28 vs. WT: Δ −0.19, p = 0.0379 and T89I: Δ11.67, p < 0.0001; −70 mV – V620I: Δ46.13 vs. WT: Δ −8.47, p < 0.0001 and T89I: Δ36.33, p = 0.0057; *Figure 1B*). Interestingly, T89I-TRPV4 was not significantly different from WT despite also causing a gain-of-function in recombinant channels (*Loukin et al., 2011*). Further, the increase in signaling in V620I only may indicate different mechanisms of action leading to the varying disease caused by the two mutations.

Next, we activated WT and mutant TRPV4 with chemical agonist GSK1016790A (GSK101) (*Jin et al., 2011*) and found that the mutations decreased the cellular response to the agonist, resulting in reduced $Ca^{2+}$ signaling. These results were supported using two methods: inside-out excised patches and confocal imaging of $Ca^{2+}$ signaling (*Figure 1C, D*). The representative traces of inside-out patches showed increased current through the patch with the addition of GSK101 and the attenuation by GSK205 (*Figure 1C*). GSK205 continued to block the channel and prevented another increase in current despite the addition of GSK101. Though the unitary currents were indistinguishable (8 pA at −30 mV) among WT and mutants, in excised inside-out patches, WT typically produced higher GSK101-induced currents than the mutants (WT: 290 pA vs. V620I: 87.1 pA and T89I: 62.3 pA at −30 mV), potentially indicative of more channels per patch (*Figure 1C*). In the confocal imaging experiments, a ratiometric fluorescence indicated $Ca^{2+}$ signaling of the hiPSC-derived chondrocytes in response to either 10 nM GSK101 or a cocktail of 10 nM GSK101 and 20 µM GSK205. WT cells had significantly higher fluorescence, and therefore $Ca^{2+}$ signaling, in response to GSK101 according to the plots and their area under the curve (WT: 1470 vs. V620I: 1114 and T89I: 1044; p < 0.0001; *Figure 1D, E*). The presence of GSK205 attenuated this response for all three lines, confirming the $Ca^{2+}$ influx was due to the TRPV4 ion channel (WT: 366 vs. V620I: 460 vs. T89I: 358). We also evaluated the response time of the cells to GSK101 and GSK101 + GSK205. We considered a cell to be responding if more than a quarter of its frames, after stimuli, had a fluorescence higher than the mean baseline plus 3 times the standard deviation. The mutants responded faster to GSK101 than the WT (WT: 46.2 s vs. V620I: 12 s, p = 0.0048 and T89I: 10.8 s, p = 0.0097; *Figure 1F*). Interestingly, the addition of GSK205 did not significantly slow the response of WT, but it did slow the response of the mutants, with the severe mutation slower than the moderate (WT: 35.4 s vs. V620I: 234 s and T89: 366 s; p < 0.0001; *Figure 1F*). These data highlight that the mutations alter the activation kinetics of TRPV4, which could play a role in the disease phenotype.

### Chondrogenic differentiation of WT and mutant hiPSC lines

To investigate if the hiPSCs with dysplasia-causing mutations exhibit altered chondrogenesis, we differentiated CRISPR-Cas9-edited hiPSCs with mutant *TRPV4* alongside an isogenic WT using our

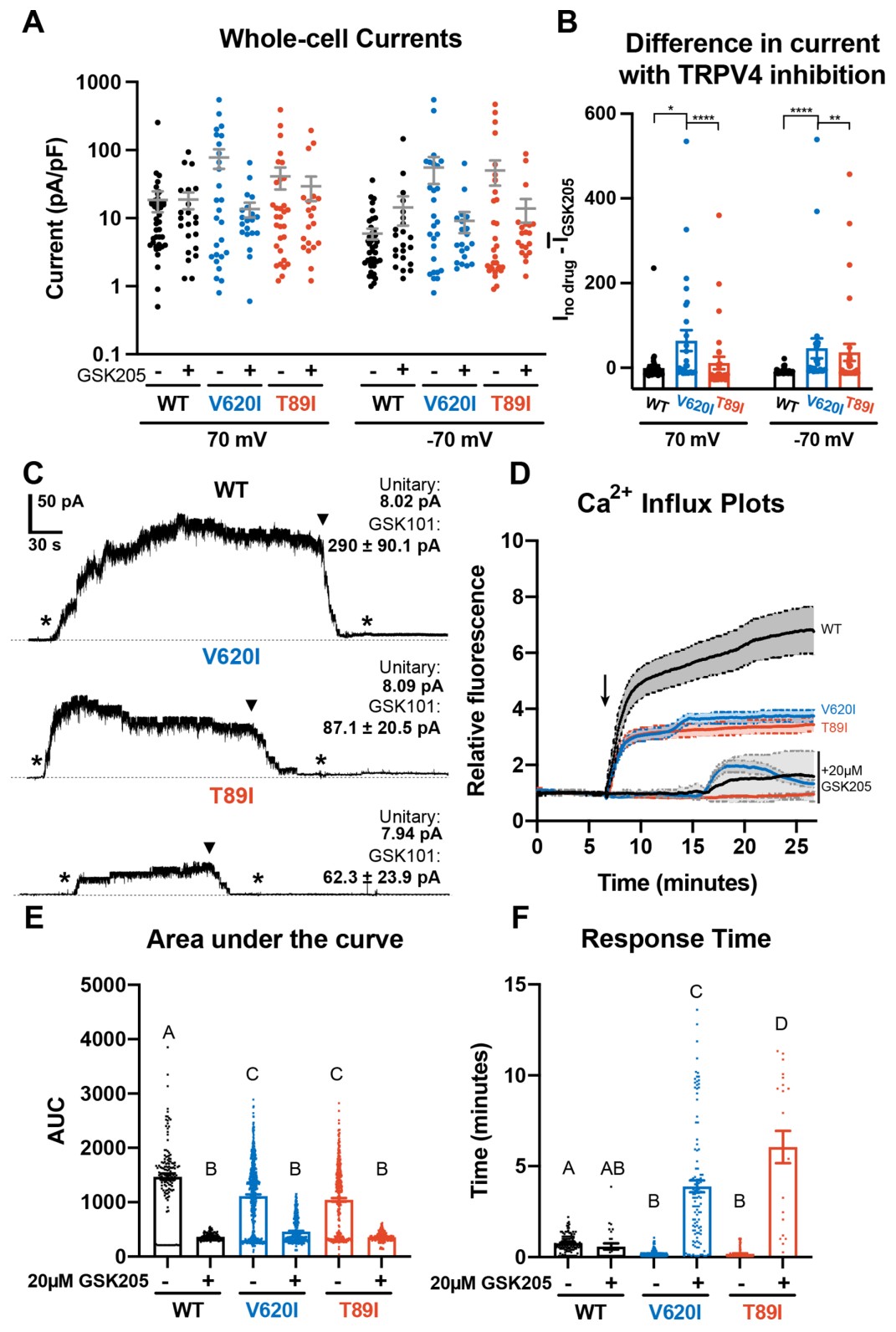

**Figure 1.** Differences in TRPV4 electrophysiological properties of wildtype (WT) and mutant human-induced pluripotent stem cell (hiPSC)-derived chondrocytes. (**A**) Whole-cell currents were higher, on average, in mutant hiPSC-derived chondrocytes than WT at 70 and −70 mV. TRPV4 inhibition with 20 μM GSK205 reduced mutant currents to similar levels as WT. Mean ± standard error of the mean (SEM). *n* = 20–40 cells from 4 differentiations.

*Figure 1 continued on next page*

*Figure 1 continued*

Kruskal-Wallis test with multiple comparisons comparing cell lines at 70 and -70 mV. No significance. (**B**) The difference between the current (*I*) through TRPV4 without GSK205 from the average current through inhibited channels was significantly higher in V620I. There was no difference between no drugs and GSK205 in WT. Mean ± SEM. *n* = 27–40 from 4 differentiations. Kruskal–Wallis test with multiple comparisons comparing cell lines at 70 and −70 mV. *p < 0.05, **p < 0.01, ****p < 0.001. (**C**) Inside-out excised patches of WT had a higher current in response to 10 nM GSK101 (indicated by *) than mutants. The addition of 10 nM GSK101 + 20 μM GSK205 (indicated by arrow head) decreased the current and continued to block the channel when GSK101 alone was re-introduced (*). Representative plots with average unitary current and current in response to GSK101. Mean ± SEM. *N* = 5, 9, and 8 for WT, V620I, and T89I, respectively, from 2 differentiations. (**D**) Mutant TRPV4 decreased the channels' sensitivity to activation with GSK101 (indicated by arrow) as shown with confocal imaging of ratiometric fluorescence indicating $Ca^{2+}$ signaling. GSK205 attenuated GSK101-mediated signaling. Mean ± 95% CI. *n* = 3 experiments with a total of 158–819 cells per line. (**E**) Quantification of the area under the curve of (**D**). Mean ± SEM. *n* = 158–819 cells from 3 experiments. Ordinary two-way analysis of variance (ANOVA) with Tukey's post hoc test. Interaction, cell line, and treatment p < 0.0001. Different letters are significantly different, p < 0.05, from each other. (**F**) Time of initial response of each responding cell (≥25% of frames for that cell are responding) measured from the addition of stimulus. Mutant TRPV4 responded faster to GSK101, but the response was significantly slowed by GSK205. Responding frames were considered to have a fluorescence greater than the mean plus three times the standard deviation. Mean ± SEM. *n* = 21–360 responding cells from 3 experiments. Ordinary two-way ANOVA with Tukey's post hoc test. Interaction, cell line, and treatment p < 0.0001.Different letters are significantly different, p < 0.05, from each other.

previously published protocol (*Adkar et al., 2019*; *Wu et al., 2021*). After 12 days of monolayer meso-dermal differentiation, the cells underwent 42 days of chondrogenic differentiation, and pellets were collected at days 7, 14, 28, and 42. Since we had previously shown that 28 days is sufficient for hiPSC chondrogenesis, day 28 was our primary time point while days 7 and 14 identified changes during differentiation. We included day 42 data in the supplement to investigate any potential changes in transcriptomic profiles and cartilaginous matrix production in chondrocyte maturation. At day 28, the three lines had similar chondrogenic matrix as shown with Safranin-O staining for sulfated glycosami-noglycans (sGAGs) and collagen type 2 alpha chain 1 (COL2A1) labeling with immunohistochemistry (IHC; *Figure 2A, B*). All three lines had little to no labeling of fibrocartilage marker COL1A1 and hypertrophic cartilage marker COL10A1 with IHC (*Figure 2C, D*). To quantitatively confirm the matrix production throughout chondrogenesis, we performed biochemical assays to measure sGAG produc-tion and normalized it to double-stranded DNA content. As expected, differences in matrix produc-tion were significant between time points (p < 0.0001; *Figure 2E*). The sGAG/DNA ratio increased in WT by 8-fold and in V620I and T89I by 5- to 5.5-fold from day 14 to 28 (p < 0.0001; *Figure 2E*). V620I pellets also increased in matrix content by 150% from day 28 to 42 (p = 0.0163; *Figure 2—figure supplement 2–1A*) with all three lines reaching an sGAG/DNA ratio of approximately 30. However, there were no differences in sGAG/DNA ratios among the three cell lines at any time point (cell line: p = 0.1206; interaction: p = 0.7426; *Figure 2E*).

Atomic force microscopy (AFM) was then used to measure the mechanical properties of the hiPSC-derived cartilaginous matrix deposited by the WT and two TRPV4-mutated cell lines. The elastic modulus ranged from 14 to 20 kPa, consistent with mouse iPSC-derived cartilage (*Diekman et al., 2012*). At day 28, the three lines had similar properties (WT: 14.4 kPa vs. V620I: 15.9 kPa vs. T89I: 14.8 kPa; *Figure 2F*); however, at day 42, V620I had a significantly decreased elastic modulus (V620I: 10.32 kPa vs. WT: 20.0 kPa, p = 0.0004 and T89I: 17.5 kPa, p = 0.0328; *Figure 2—figure supplement 2–1B*). These experiments indicated that all three lines properly differentiated into chondrocytes and had similar cartilaginous matrix production at day 28. With 14 more days of chondrogenic culture, minor differences in matrix accumulation were observed with the moderate V620I line.

## TRPV4 mutations altered chondrogenic gene expression in hiPSC-derived chondrocytes

Reverse transcription quantitative polymerase chain reaction (RT-qPCR) analysis throughout differ-entiation shows that mutants had higher *ACAN* expression compared to WT at day 28 (day-28 fold changes; WT: 2314 vs. V620I: 6418, p = 0.1092 and T89I: 5870, p = 0.0316; *Figure 3A*); however, expression decreased at day 42 in T89I (*Figure 3—figure supplement 1A*). *COL2A1* expression was

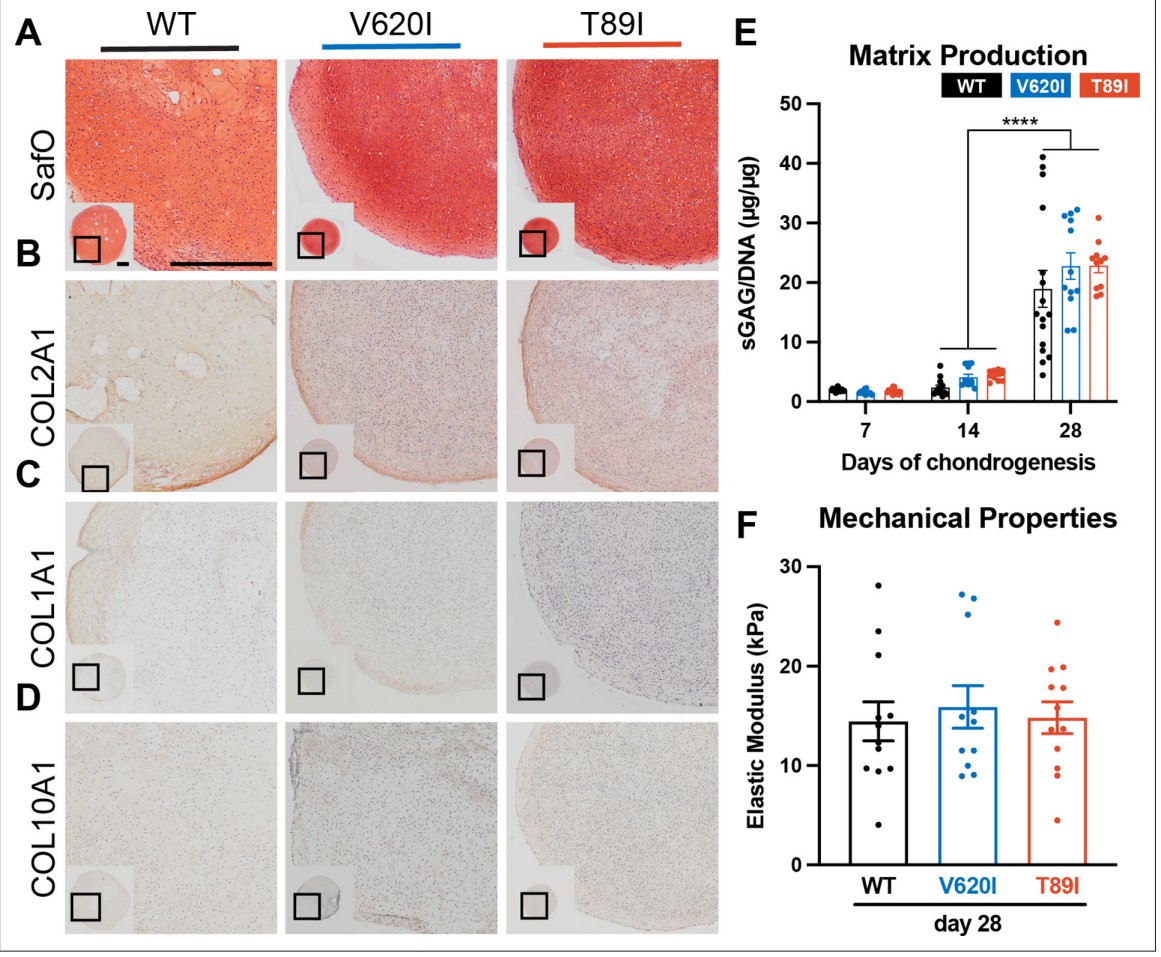

**Figure 2.** Mutant TRPV4 had little effect on chondrogenic matrix production. (**A**) Wildtype (WT), V620I, and T89I day-28 pellets exhibit similar matrix production shown by staining for sulfated glycosaminoglycans (sGAGs) with Safranin-O and hematoxylin and labeling with immunohistochemistry (IHC) for (**B**) COL2A1 (**C**), COL1A1 (**D**), and COL10A1. Scale bar = 500 μm. Representative images from 3 to 4 differentiations. (**E**) The sGAG/DNA ratio increased in all three lines from day 14 to 28 of chondrogenesis. There were no differences between lines at each time point. Mean ± standard error of the mean (SEM). $n$ = 11–16 from 3 to 4 independent differentiation experiments. ****$p < 0.0001$ Statistical significance determined by an ordinary two-way analysis of variance (ANOVA) with Tukey's post hoc test. (**F**) There were no differences in the elastic modulus of the matrix at day 28. Mean ± SEM. $n$ = 11–14 from 3 experiments. Statistical significance determined by an ordinary two-way ANOVA with Tukey's post hoc test.

The online version of this article includes the following figure supplement(s) for figure 2:

**Figure supplement 1.** Minor differences in V620I matrix were observed at day 42 of chondrogenesis.

similar among the three lines at day 28 (day-28 fold changes; WT: 6492 vs. V620I: 6524, p > 0.9999 and T89I: 8131, p = 0.3304; *Figure 3B*) but significantly lower in T89I at day 42 (day-42 fold changes; T89I: 2798 vs. WT: 9209, p = 0.0144 and V620I: 7177, p = 0.0007; *Figure 3—figure supplement 1B*). Throughout chondrogenesis, V620I significantly increased expression of chondrogenic transcription factor *SOX9* (day-28 fold changes; V620I: 178.9 vs. WT: 49.16, p = 0.0011 and T89I: 55.37, p = 0.0117; *Figure 3C*) and *TRPV4* (day-28 fold changes; V620I: 112.1 vs. WT: 42.14, p < 0.0001 and T89I: 45.82, p = 0.0002; *Figure 3D*). On the other hand, T89I significantly increased expression of pro-inflammatory, calcium-binding protein *S100B* (*Yammani, 2012*) throughout chondrogenesis (day-28 fold changes; T89I: 2552 vs. WT: 415.8, p < 0.0001 and V620I: 633.6, p = 0.0019; *Figure 3E*). T89I also had significantly higher expression of fibrocartilage marker *COL1A1* at days 7, 14, and 28 than the other two lines (day-28 fold changes; T89I: 80.33 vs. WT: 13.15, p < 0.0001 and V620I: 23.61, p = 0.0043; *Figure 3F*), and both mutations had increased expression at day 42 compared to WT (*Figure 3—figure supplement 1F*). In contrast, hypertrophic marker *COL10A1* was significantly higher in the WT line than the mutants at days 28 and 42 (day-28 fold changes; WT: 310.4 vs. V620I: 30.26, p = 0.0338 and T89I: 52.92, p = 0.0033; *Figure 3G*; *Figure 3—figure supplement 1G*). Surprisingly, there was no

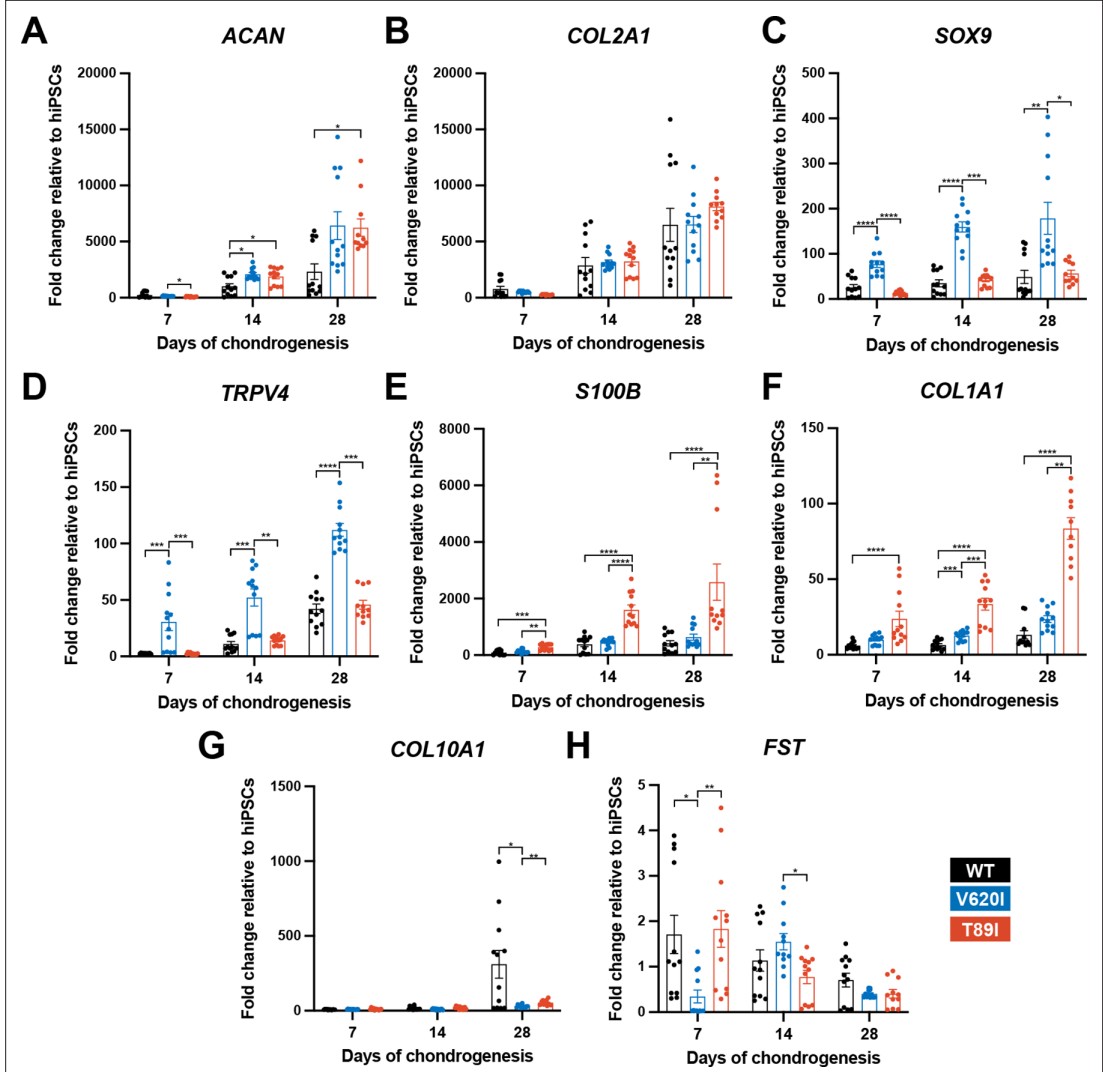

**Figure 3.** V620I and T89I exhibited differing effects on gene expression during chondrogenic differentiation. (**A**) V620I and T89I had increased *ACAN* gene expression at day 28 compared to wildtype (WT). (**B**) The three lines had similar *COL2A1* expression throughout differentiation. V620I increased expression of (**C**) *SOX9* and (**D**) *TRPV4* throughout chondrogenesis. T89I increased expression of (**E**) *S100B* and (**F**) *COL1A1* throughout chondrogenesis. (**G**) Both mutations decreased *COL10A1* gene expression at day 28 compared to WT. (**H**) There were no differences in *FST* expression at day 28. Mean ± standard error of the mean (SEM). *n* = 10–12 from 3 independent differentiation experiments. *p < 0.05, **p < 0.01, ***p < 0.001, ****p < 0.0001 Significance determined by one-way analysis of variance (ANOVA) with Tukey's post hoc test for each time point.

The online version of this article includes the following figure supplement(s) for figure 3:

**Figure supplement 1.** V620I and T89I had differing effects on gene expression during chondrogenic differentiation.

significant increase in follistatin (*FST*) expression in mutants at later time points (day-28 fold changes; WT: 0.7042 vs. V620I: 0.4025, p = 0.6228 and T89I: 0.4242, p > 0.9999; *Figure 3H*) despite previous findings (*Leddy et al., 2014b*).

To obtain comprehensive transcriptomic profiles of WT and TRPV4-mutated cell lines, we performed bulk RNA sequencing of day-28 chondrogenic pellets. We compared V620I and T89I gene expression to WT and plotted the log$_2$ fold change in heatmaps (*Figure 4A, B*). While many chondrogenic and hypertrophic genes had similar levels of expression between the lines, the mutants had increased expression of cartilage oligomeric matrix protein (*COMP*), collagen type VI alpha chains 1 and 3 (*COL6A1, COL6A3*), growth differentiation factor 5 (*GDF5*), high-temperature requirement A serine peptidase 1 (*HTRA1*), and secreted protein acidic and cysteine rich (*SPARC*) (*Figure 4A*). Additionally, the mutations up-regulated expression levels of bone morphogenic protein 6 (*BMP6*), transforming growth factor 3 (*TGFB3*), nuclear factor of activated T-Cells C2 (*NFATC2*), Twist family

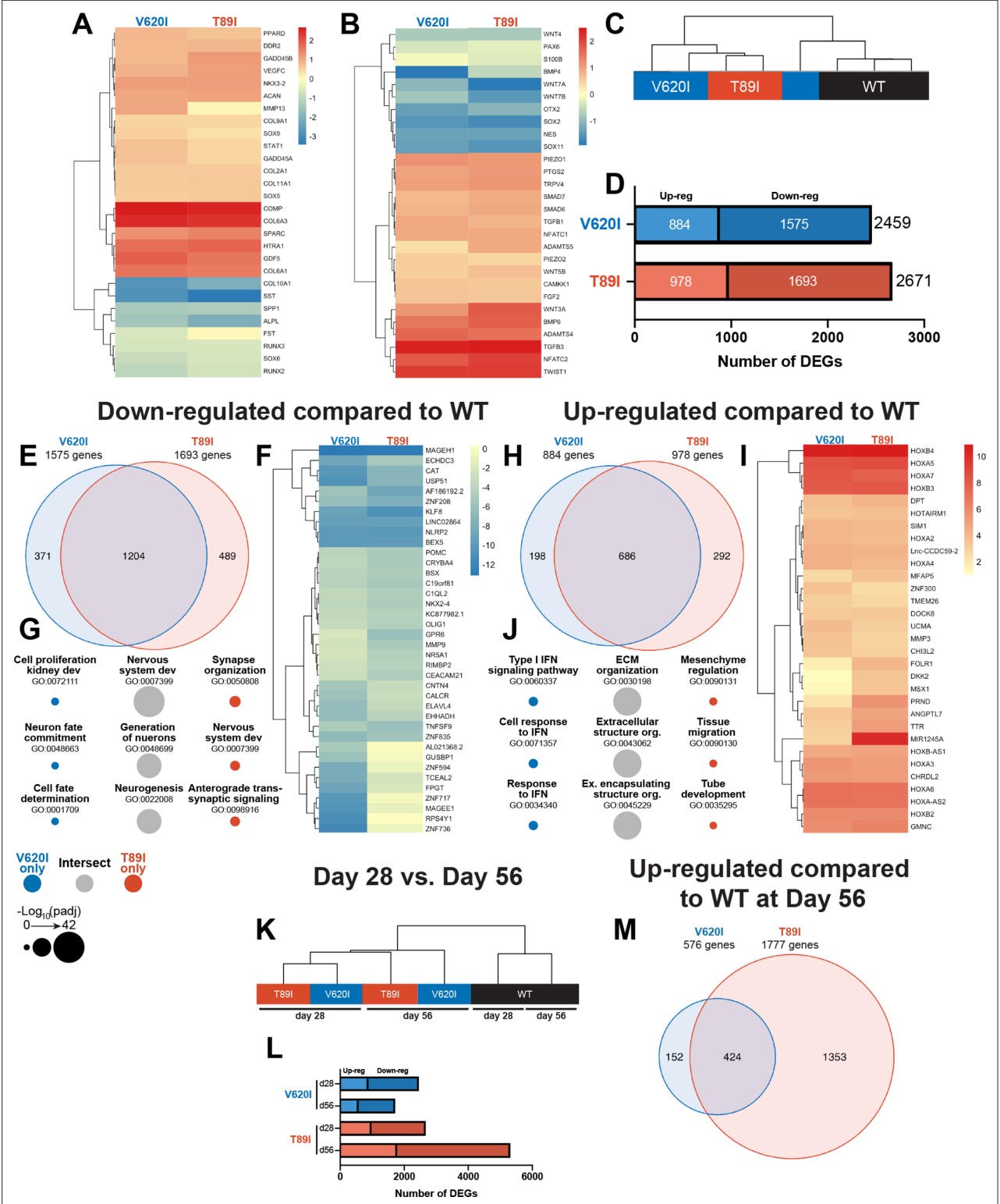

**Figure 4.** Dynamic changes in transcriptomic profiles of V620I and T89I mutants during chondrogenesis. Heatmaps comparing the log₂ fold change of common chondrogenic and hypertrophic genes (**A**) and growth factor and signaling genes (**B**) in day-28 V620I and T89I chondrocytes compared to wildtype (WT). (**C**) Clustering of the samples using Euclidean distances reveals that V620I and T89I human-induced pluripotent stem cell (hiPSC)-derived chondrocytes are more similar to each other than WT. (**D**) The number of up- and down-regulated differentially expressed genes (DEGs) in V620I and T89I day-28 chondrocytes compared to WT. (**E–G**) Analysis of the down-regulated genes compared to WT. (**E**) A Venn diagram reveals the number of similar and different down-regulated DEGs between V620I and T89I, where most genes are shared. (**F**) A heatmap showing the log₂ fold change,

*Figure 4 continued on next page*

*Figure 4 continued*

compared to WT, of the top 25 down-regulated genes for each line. (**G**) The top 3 Gene Ontology (GO) terms (biological process) associated with the DEGs unique to V620I, shared between V620I and T89I, and unique to T89I. Symbol color represents the cell line, and size represents the $-\log_{10}(p_{adj})$. (**H–J**) Analysis of the up-regulated genes compared to WT. (**H**) A Venn diagram reveals the number of similar and different up-regulated DEGs between V620I and T89I, where most genes are shared. (**I**) A heatmap showing the $\log_2$ fold change, compared to WT, of the top 25 up-regulated genes for each line. (**J**) The top 3 GO terms (biological process) associated with the DEGs unique to V620I, shared between V620I and T89I, and unique to T89I. Symbol color represents the cell line, and size represents the $-\log_{10}(p_{adj})$. (**K**) Clustering of the day-28 and -56 samples using Euclidean distances reveals that the WT chondrocytes, at both days 28 and 56, cluster together while mutants cluster by time point. (**L**) The number of up- and down-regulated DEGs for V620I and T89I compared to WT at days 28 and 56. (**M**) A Venn diagram reveals the number of similar and different up-regulated DEGs between V620I and T89I, with T89I becoming more unique at day 56. *n* = 3–4 samples.

The online version of this article includes the following figure supplement(s) for figure 4:

**Figure supplement 1.** Distinction between V620I and T89I.

**Figure supplement 2.** Top differentially expressed genes (DEGs) of V620I and T89I chondrocytes compared to wildtype (WT) remain from day 28 to 56.

BHLH transcription factor 1 (*TWIST1*), ADAM metallopeptidase with thrombospondin type 1 motif 4 (*ADAMTS4*), and *WNT3A* (*Figure 4B*). In contrast, the mutants had decreased expression of hypertrophic markers *COL10A1*, secreted phosphoprotein 1 (*SPP1*), and alkaline phosphatase, biomineralization associated (*ALPL*) (*Figure 4B*). The mutations also down-regulated osteoblastogenesis transcription factors *SOX2* and *SOX11* and previously identified genes governing off-target differentiation during hiPSC chondrogenesis including nestin (*NES*), orthodenticle homeobox 2 (*OTX2*), *WNT7A*, and *WNT7B* (*Figure 4B*). Overall, these results indicate the mutant chondrocytes express higher levels of chondrogenic markers and lower levels of genes associated with hypertrophy compared to WT.

## V620I and T89I mutants demonstrate similar gene expression profiles early in differentiation

First, to evaluate the similarities and differences in transcriptomic profiles between the hiPSC-derived chondrocytes with and without the TRPV4 mutations, we computed the Euclidean distance between day-28 samples of each cell line. The WT samples clustered away from the mutants, and the V620I samples were the most variable. (*Figure 4C*). In terms of total differentially expressed genes (DEGs) compared to WT, V620I had 8% fewer DEGs than T89I (2459 vs. 2671; *Figure 4D*). Mutants had only about half of the number of up-regulated genes compared to down-regulated genes (V620I: 884 vs. 1575, T89I: 978 vs. 1693; *Figure 4D*). The majority of the down-regulated DEGs were shared between the two mutants when compared to WT, comprising 76% and 71% of V620I's and T89I's total down-regulated DEGs, respectively (*Figure 4E*). We plotted the top 25 most down-regulated DEGs for each line in a heatmap. These included antioxidant catalase (*CAT*), anti-inflammatory nucleotide-binding and leucine-rich repeat receptor family pyrin domain containing 2 (*NLRP2*), and Kruppel-like factor 8 (*KLF8*) (*Figure 4F*). Interestingly, many of the down-regulated DEGs, both unique and shared between V620I and T89I, were associated with Gene Ontology (GO) terms related to nervous system development, including many potassium channel genes (i.e., *KCN* family; *Figure 4G*). This finding is potentially indicative of changes in ion channel signaling beyond TPRV4 with the mutations.

In contrast, 686 up-regulated DEGs were shared by both mutants, while 22% of V620I's and 30% of T89I's up-regulated DEGs were unique to each mutation (198 vs. 292; *Figure 4H*). A heatmap of the top 25 up-regulated DEGs showed that several homeobox (HOX) genes were highly expressed in chondrocytes with the TRPV4 mutations (*Figure 4I*). These included *HOXA2* to *HOXA7*, *HOXA-AS2*, *HOXB2* to *HOXB4*, and *HOXB-AS1*, which are associated with morphogenesis and anterior patterning (*Seifert et al., 2015*). Furthermore, the shared, up-regulated DEGs between two mutants are associated with extracellular matrix production and organization and growth factor binding in GO term analysis, while V620I genes were associated with type I interferon (*Figure 4J*). These data highlighted an early morphogenic genetic profile in hiPSC-derived chondrocytes with the V620I and T89I mutations.

Additionally, while mutated chondrocytes were more similar to each other compared to WT, we identified a set of genes that may regulate the different disease phenotypes of moderate brachyolmia and severe metatropic dysplasia caused by the V620I and T89I mutation, respectively. For example, the top 15 up- and down-regulated genes unique to either V620I or T89I were plotted in a heatmap (*Figure 4—figure supplement 1A*). Interferon-induced protein with tetratricopeptide repeats 3 (*IFIT3*), interferon-induced GTP-binding protein Mx1 (*MX1*), and p53 up-regulated regulator of p53

levels (*PURPL*) were all up-regulated in V620I, but not T89I, consistent with the associated pathways regarding interferon signaling (*Figure 4—figure supplement 1B* and *Figure 4J*). Interestingly interferonopathies with enhanced type 1 signaling may lead to intracranial calcification and skeletal development problems (*Yu and Song, 2020*). We also observed that protein kinase C alpha (PKC; *PRKCA*), which plays a role in the phosphorylation of TRPV4, was up-regulated in V620I compared to WT (*Figure 4—figure supplement 1B*). V620I also uniquely had many down-regulated genes related to DNA- and RNA-binding such as zinc finger proteins (*ZNF736*, *ZNF717*, and *ZNF594*) and ribosomal protein S4 y-linked 1 (*RPS4Y1*; *Figure 4—figure supplement 1A*). T89I had much higher expression of micro-RNA *MIR1245A*, compared to both WT and V620I, which has been shown to increase proliferation in colon cancer (*Pan et al., 2019*; *Figure 4—figure supplement 1A*). Developmental protein dickkopf WNT signaling pathway inhibitor 2 (*DKK2*) and carbonic anhydrase 2 (*CA2*), which is essential for bone resorption, were also uniquely up-regulated in T89I at day 28 (*Figure 4—figure supplement 1A*). In contrast, T89I had reduced expression of bone matrix structural protein bone sialoprotein II (*IBSP*) and limb development transcription factor *SP9* (*Figure 4—figure supplement 1A*). These mutant-specific DEGs highlight that the severe T89I mutation began to have a unique skeletal development transcriptome as early as day 28.

## The severe T89I mutation inhibits chondrocyte hypertrophy more than moderate V620I mutation

Following an additional 4 weeks of chondrogenic culture, we performed RNA sequencing to investigate how the differences between the WT and the two mutants change with further differentiation. Using Euclidean distances, we compared the WT, V620I, and T89I hiPSC-derived chondrocytes at both days 28 and 56 (*Figure 4K*). WT clustered together at both days 28 and 56; however, the mutants clustered by time point. Again, there were more down-regulated genes than up-regulated at day 56 (*Figure 4L*). The lethal, metatropic-dysplasia-causing T89I mutation had the most DEGs, and the number increased from day 28 to 56. In contrast, the moderate, brachyolmia-causing V620I mutation DEGs decreased at day 56. 74% of V620I up-regulated DEGs, but only 24% of T89I DEGs, were shared between the two lines (424 total genes; *Figure 4M*). These intersecting, up-regulated genes were associated with the biological processes of skeletal development, morphogenesis, and patterning due to the up-regulation of many *HOX* genes (*Figure 4—figure supplement 2A*). Most of the top up- and down-regulated genes were consistent between days 28 and 56 (*Figure 4—figure supplement 2A–B*), including both anterior and posterior *HOX* genes (i.e., *HOXA1* to *HOXA7*, *HOXB2* to *HOXB4*, *HOXB6* to *HOXB8*, *HOXC4*, *HOXD8*, *HOXA-AS2-3*, and *HOXB-AS1-2*) (*Seifert et al., 2015*). Although V620I and T89I TRPV4 mutants continued to share the up-regulated *HOX* genes, which may be responsible for dysfunctional chondrogenic hypertrophy compared to WT cells, our results also indicate that these two mutated lines started to demonstrate further divergent transcriptomic profiles in later chondrogenesis. We observed more up- and down-regulated DEGs in T89I vs. WT compared to V620I vs. WT. Insulin growth factor-like family member 3 (*IGFFL3*) and matrix extracellular phosphoglycoprotein (*MEPE*) were significantly up-regulated in T89I; however, they were slightly down-regulated in V620I, compared to WT (*Figure 4—figure supplement 1C*). T89I also up-regulated calcium-binding proteins annexin A8 (*ANXA8*) and *S100A3* (*Figure 4—figure supplement 1C*). Consistent with its regulation of many Wnt-related genes, T89I up-regulated beta catenin (*CTNNB1*) compared to WT at day 56 (*Figure 4—figure supplement 1D*). Both T89I and V620I uniquely down-regulated DNA- and RNA-binding genes, such as various zinc finger proteins, with many of same up- and down-regulated genes in V620I at day 56 as 28 (*Figure 4—figure supplement 1C*). Moderate V620I's difference from WT remained, while T89I continued to diverge with further differentiation.

## TRPV4 mutations exhibit dysregulated BMP4-induced chondrocyte hypertrophy

To evaluate how TRPV4 mutations may affect hypertrophy, BMP4 was added to the chondrogenic medium with and without TGFβ3 to stimulate hypertrophic differentiation starting at day 28 of chondrogenic pellet culture (*Craft et al., 2015*). At day 56, Safranin-O staining indicated the BMP4-treated WT had developed a more hypertrophic phenotype compared to TGFβ3- and TGFβ3 + BMP4-treated pellets with enlarged chondrocytes (cell diameter; WT-BMP4: 27.6 µm vs. WT-TGFβ3: 11.8 µm, V620I-BMP4: 12.5 µm, and T89I-BMP4: 11.3 µm; p < 0.0001; *Figure 5A, B*). This phenotype was not

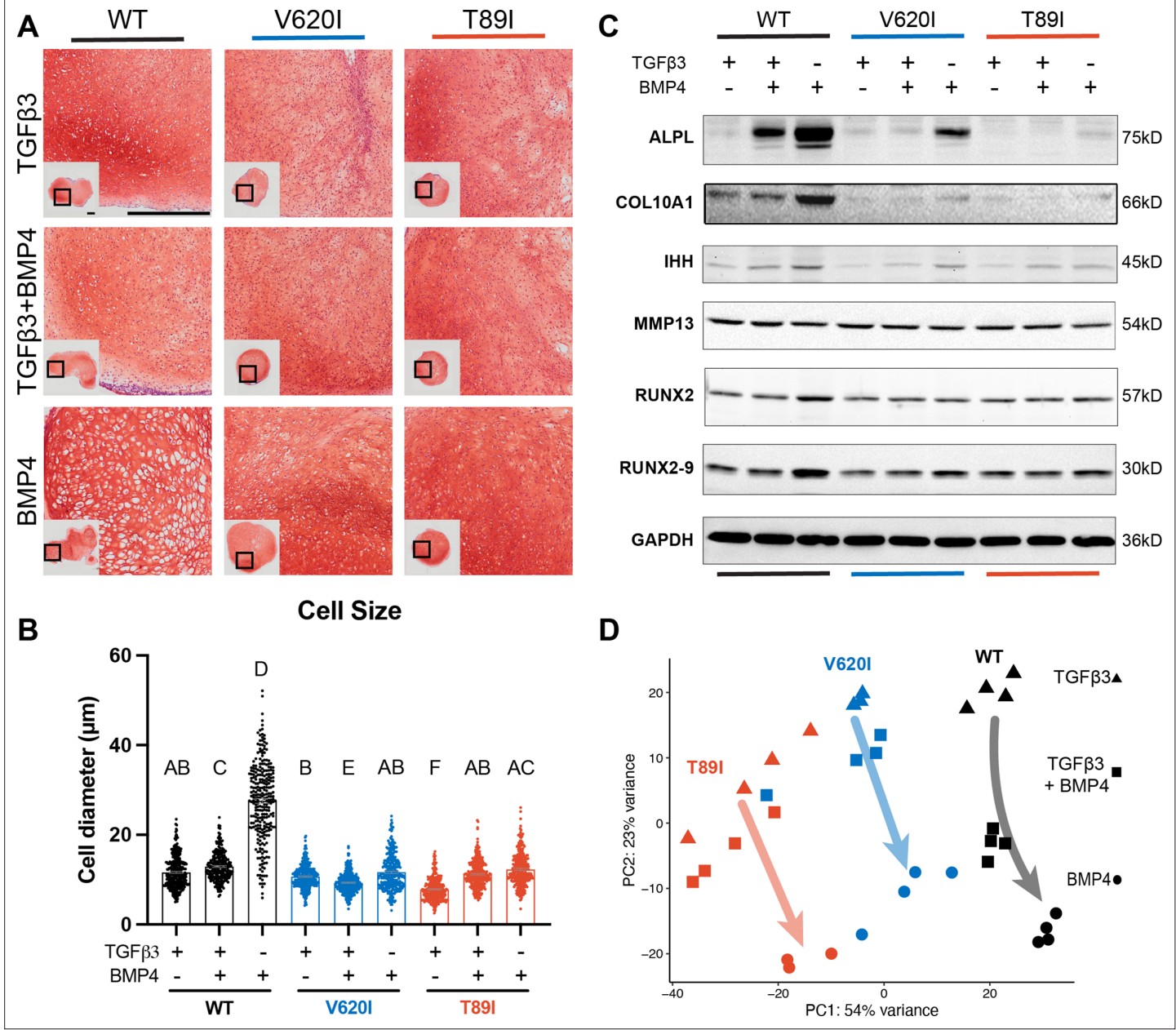

**Figure 5.** Wildtype (WT) chondrocytes are more sensitive to BMP4 treatment. (**A**) WT chondrocytes treated with BMP4 developed a hypertrophic phenotype with enlarged lacunae, which was not present in the mutant cell lines or other conditions, as shown by Safranin-O and hematoxylin staining. Scale bar = 500 µm. Representative images from 2 experiments. (**B**) Cell diameter was significantly increased in the WT with BMP4 treatment compared to all other groups indicating a hypertrophic phenotype. Mean ± standard error of the mean (SEM). $n$ = 249–304 cells from 2 pellets. Different letters indicate statistical significance ($p < 0.05$) between groups as determined by Kruskal–Wallis test with multiple comparisons since data was not normally distributed. (**C**) Western blot shows that WT had a stronger increased production of ALPL, COL10A1, IHH, RUNX2, and RUNX2-9 in response to BMP4 treatment than the mutants. (**D**) Principle component analysis (PCA) of bulk RNA-seq reveals an increased sensitivity to BMP4 (and TGFβ3 + BMP4) treatment in WT human-induced pluripotent stem cell (hiPSC)-derived chondrocytes compared to V620I and T89I. $n$ = 3–4 samples.

The online version of this article includes the following source data and figure supplement(s) for figure 5:

**Figure supplement 1.** Hypertrophic gene and protein expression.

**Source data 1.** ALPL western blot: the full raw unedited gel with and without the bands labeled.

**Source data 2.** COL10A1 western blot: the full raw unedited gel with and without the bands labeled.

**Source data 3.** IHH western blot: the full raw unedited gel with and without the bands labeled.

**Source data 4.** MMP13 western blot: the full raw unedited gel with and without the bands labeled.

*Figure 5 continued on next page*

*Figure 5 continued*
**Source data 5.** RUNX2 western blot: the full raw unedited gel with and without the bands labeled.
**Source data 6.** GAPDH western blot: the full raw unedited gel with and without the bands labeled.

present in any of the groups from the V620I and T89I lines. Western blot and RNA sequencing further confirmed BMP4-induced hypertrophy was more prominent in the WT line. There was an increase in gene expression and protein production of hypertrophic cartilage markers COL10A1, ALPL, IHH, RUNX2 isoform 9 (RUNX2-9) in all three lines with BMP4 treatment; however, there was a stronger effect in WT (*Figure 5C*; *Figure 5—figure supplement 1*). Additionally, only BMP4-treated WT had an increase in RUNX2 (*Figure 5C*; *Figure 5—figure supplement 1*). These data also highlight the stratification between the moderate V620I and severe T89I mutations as BMP4-treated T89I had lower expression and production of COL10A1, ALPL, and IHH compared to BMP4-treated V620I (*Figure 5C*; *Figure 5—figure supplement 1*). Interestingly, BMP4 treatment reduced MMP13 in the mutants but did not affect WT (*Figure 5C*; *Figure 5—figure supplement 1*). A principle component analysis (PCA) of the RNA sequencing data revealed that the WT line was overall more sensitive to BMP4, as indicated by the arrows (*Figure 5D*). Given that the BMP4-treated WT chondrocytes had the most apparent hypertrophic phenotype, later analyses were performed comparing the BMP4- and TGFβ3-treated chondrocytes for simplification.

Hierarchical *k*-means clustering of gene expression profiles of BMP4- and TGFβ3-treated chondrocytes resulted in 9 unique clusters, as determined using the gap statistics method (*Figure 6A*). Most of the clusters, including the largest (i.e., cluster 1), showed up-regulation of gene expression with BMP4 treatment, while clusters 4, 5, and 9 showed down-regulation. The gene expression per group for each cluster is listed in *Supplementary file 1*. Overall, WT responded to BMP4 treatment with the largest number of DEGs, over 2500, with only 22% of them shared among all three lines (*Figure 6B*). Although cluster 1 shows an overall increase in gene expression with BMP4 treatment, WT had a larger increase in expression than the mutants (*Figure 6C*). In fact, some of the genes that were up-regulated with BMP4 treatment in WT may have no change or down-regulation in mutants (cluster 1, *Figure 6A*).

As cluster 1 represents the primary response to BMP4 treatment and may highlight how the TRPV4 mutations inhibit chondrocyte hypertrophy, we constructed a gene network of this cluster (*Figure 6D*). The log fold change of each gene per cell line is represented by a color scale, which is consistent with WT having overall higher expression of the genes (as indicated by the white arrows in the legend; *Figure 6D*). With GO term analysis, the cluster 1 gene network is highly associated with ossification, biomineral tissue development, skeletal system development, tissue development, and osteoblast differentiation (*Figure 6D*). Alkaline phosphatase, biomineralization associated (*ALPL*), amelogenin X-linked (*AMELX*), fibroblast growth factor receptor 3 (*FGFR3*), interferon-induced transmembrane protein 5 (*IFITM5*), Indian hedgehog (*IHH*), parathyroid hormone 1 receptor (*PTH1R*), and noggin (*NOG*) were connected to at least 4 of the top 5 GO terms. Of those, *ALPL*, *AMELX*, and *IFITM5* showed much higher expression in WT than the mutants alongside antioxidant glutathione *S*-transferase alpha 1 (*GSTA1*) and bone ECM proteins integrin-binding sialoprotein (*IBSP*) and matrix extracellular phosphoglycoprotein (*MEPE*). Lack of expression of these key genes, particularly *ALPL*, may be responsible for the inhibited hypertrophy in TRPV4 V620I- and T89I-mutated chondrocytes.

We next investigated and plotted the top 25 up-regulated genes for each line with BMP4 treatment (compared to their respective TGFβ3 control) (*Figure 6E*). 88% of these genes were also present in cluster 1. The key genes *ALPL*, *AMELX*, *IFITM5*, *GSTAI*, *IBSP*, and *MEPE* had distinctly higher expression in WT than mutants, in agreement with the network analysis. Both mutants showed higher expression than WT of ankyrin repeat and SOCS box containing 10 (*ASB10*), GTPase, IMAP family member 6 (*GIMAP6*), and adhesion G-protein-coupled receptor D1 (*ADGRD1*) when compared to their corresponding TGFβ3 control group. GO term analysis was further performed on all BMP4 up-regulated DEGs for each line (*Figure 6F*). WT was highly associated with skeletal system development, ossification, endochondral ossification, and extracellular structure organization. V620I was also associated with these concepts to a lesser degree, while T89I showed little to no association. We believe these results highlight that the TRPV4 mutations reduce BMP4-induced hypertrophy but to a greater extent with the T89I mutation, which causes the more severe phenotype.

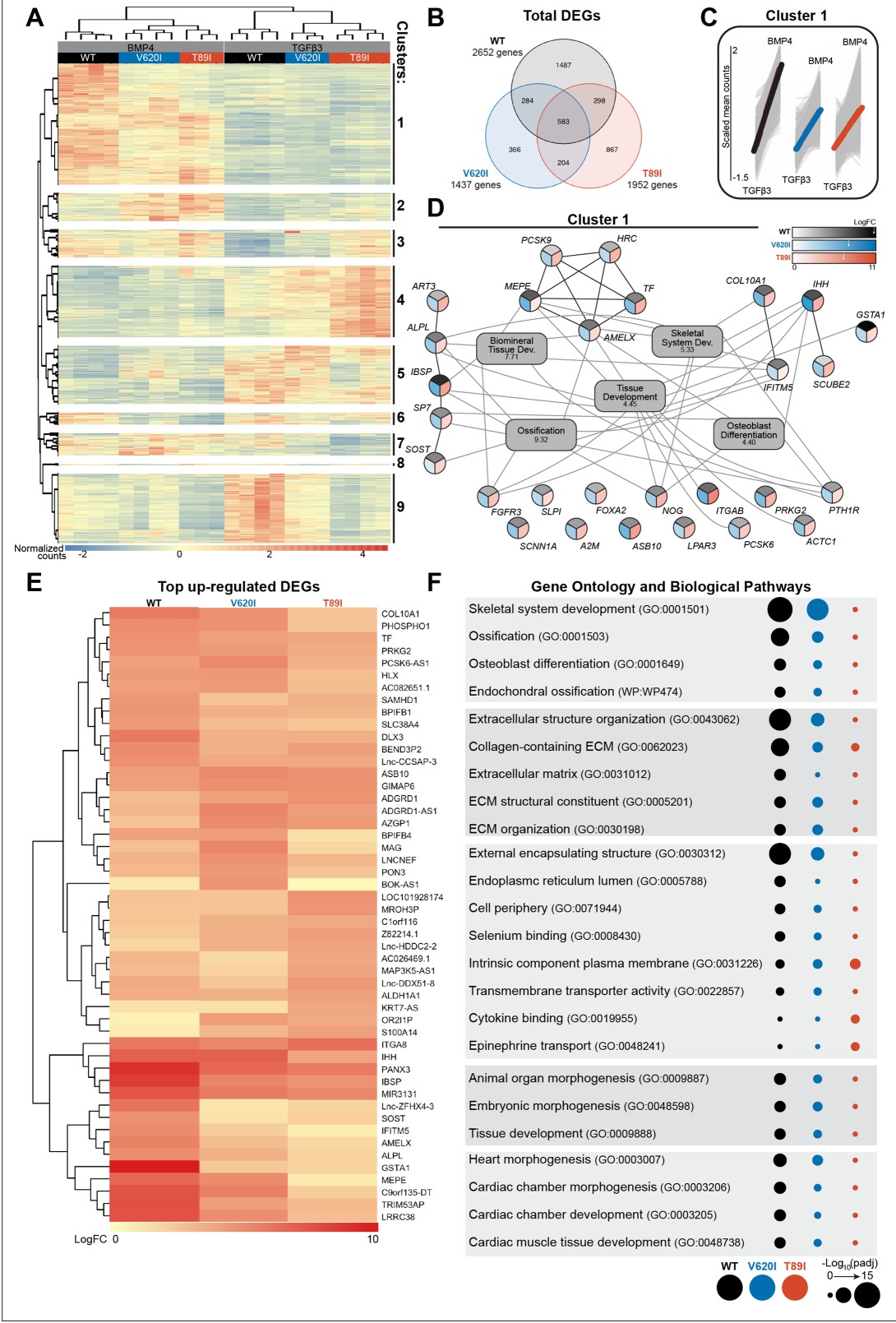

**Figure 6.** V620I and T89I had an inhibited hypertrophic response to BMP4 treatment. (**A**) There are 9 clusters of genes based on expression and hierarchical *k*-means clustering of the samples. (**B**) Venn diagram shows similar and distinct differentially expressed genes (DEGs) in response to BMP4 treatment in all three lines. (**C**) Cluster 1 represented increasing in expression from TGFβ3 to BMP4 treatment (left to right on *x*-axis). *Y*-axis scale (−1.5 to 2) represents the scaled mean counts. (**D**) A protein–protein interaction network with functional enrichment analysis of cluster 1 reveals the top

*Figure 6 continued on next page*

*Figure 6 continued*

regulating genes and their associated concepts. Connections between protein-coding genes and Gene Ontology (GO) processes are based on the average log fold change between cell lines. Coloring of the protein-coding gene circles is divided into three to represent the log fold change for each cell line as shown in the legend. The white arrows in the legend indicates the location of the maximum log fold change for each respective cell line. The gray boxes represent the top 5 GO terms (biological process) identified for the network with the $\log_{10}$(false discovery rate) underneath the term. (**E**) A heatmap of the top 25 up-regulated genes, and their $\log_2$ fold change, in each line compared to their respective TGFβ3 controls. (**F**) The top GO terms and biological pathways associated with the up-regulated DEGs with BMP4 treatment. Symbol color represents the cell line, and size represents the $-\log_{10}(p_{adj})$.

## Discussion

To elucidate the detailed molecular mechanisms underlying the distinct severity of skeletal dysplasias caused by two TRPV4 mutations (moderate brachyolmia-causing V620I vs. severe metatropic dysplasia-causing T89I), we used CRISPR-Cas9 gene editing to generate hiPSC-derived chondrocytes bearing either V620I or T89I mutation. We observed that day-28 chondrocytes exhibited differences in channel function and gene expression between the mutants and WT control. Differences in transcriptomic profiles between V620I and T89I and from WT became more apparent with maturation following 4 additional weeks of culture with TGFβ3 or hypertrophic differentiation with BMP4 treatment. Of note, WT was significantly more sensitive to BMP4-induced hypertrophy. At the transcriptomic and proteomic levels, TRPV4 mutations inhibited chondrocyte hypertrophy, particularly with the T89I mutation, whereas V620I exhibited a milder phenotype, consistent with the clinical presentation of these two conditions. Our results suggest that skeletal dysplasias may be, at least in part, resulting from improper chondrocyte hypertrophy downstream of altered TRPV4 function. Furthermore, with our genome-wide RNA sequencing analysis, we also identified several putative genes that may be responsible for these dysregulated pathways in human chondrocytes bearing V620I or T89I TRPV4 mutations.

Our findings are generally consistent with previous non-human models of V620I and T89I mutations. Two other models that have studied the V620I and T89I mutations include *X. laevis* oocytes injected with rat TRPV4 cRNA (*Loukin et al., 2011*) or primary porcine chondrocytes transfected with human mutant TRPV4 (*Leddy et al., 2014b*). Both reports and our current study investigated the baseline currents of the mutant TRPV4 compared to WT. Here, we used patch clamping and observed high basal currents in V620I with a significant decrease when TRPV4 was inhibited. However, this characteristic was trending, but not significant, in T89I, despite both V620I and T89I being reported as gain-of-function mutations (*Camacho et al., 2010*; *Rock et al., 2008*). Both the *X. laevis* oocyte and porcine chondrocyte models confirmed high basal currents through V620I-TRPV4 (*Leddy et al., 2014b*; *Loukin et al., 2011*). Interestingly, *X. laevis* oocytes, but not the humanized porcine chondrocytes, showed an increase in basal Ca²⁺ signaling through T89I (*Leddy et al., 2014b*; *Loukin et al., 2011*). Furthermore, our results were consistent with a summary of TRPV4 channelopathies reporting an increase in conductivity in V620I but no change in T89I (*Kang, 2012*). Interestingly, V620I also had increased expression of *PRKCA*, the gene encoding for protein kinase C alpha. Phosphorylation by PKC has been shown to alter TRPV4 activation (*Cao et al., 2018*) and therefore may play a role in the altered signaling with these mutations. In future experiments, we will further investigate PKC and PKA phosphorylation of TRPV4 and the effects on channel activity in these mutations. The conflicting basal current results could be due to differences in phosphorylation or the species of the TRPV4, but this was not the case regarding channel activation. As mentioned, the hiPSC-derived chondrocytes with V620I and T89I TRPV4 had reduced currents and Ca²⁺ signaling in response to chemical agonist GSK101. However, our previous study showed the porcine chondrocytes with mutant human TRPV4 had increased peak Ca²⁺ signaling in response to hypotonic changes (*Leddy et al., 2014b*). This discrepancy could be due to the mode of activation of TRPV4 (i.e., osmotic vs. chemical agonist). In contrast, the oocytes with mutant rat TRPV4 had lower currents in response to both hypotonic and chemical (GSK101) TRPV4 activation compared to WT-TRPV4, consistent with our findings. It can be speculated that there is decreased sensitivity to the antagonist because the mutated hiPSC-derived chondrocytes are compensating for the increased basal activity by reducing the number of TRPV4 channels or other ion channels and signaling transducers as shown with the RNAseq data and associated GO terms. The increased basal currents and decreased channel sensitivity to TRPV4 agonist GSK101 with mutated TRPV4 are also likely resulting from an increased open probability of TRPV4

making the channels less likely to be activated by a chemical agonist (*Loukin et al., 2011*). The obvious differences in both resting and activated states confirm functional differences with TRPV4 mutations that may ultimately lead to changes in downstream signaling of the channel, which alter joint development and result in skeletal dysplasias.

It was hypothesized, in the porcine chondrocyte study, that the increased $Ca^{2+}$ signaling due to the V620I and T89I TRPV4 mutations increased *FST* expression that inhibited BMP signaling and hypertrophy (*Leddy et al., 2014a*; *Leddy et al., 2014b*). Surprisingly, we found no differences in *FST* expression in mutant hiPSC-derived chondrocytes compared to WT. However, our previous study used non-human cells, which could alter the effects of the human TRPV4 mutations and downstream gene expression. Another previous hypothesis made was that the altered TRPV4 signaling increased *SOX9* expression, a known regulator of resting and proliferating chondrocytes up-regulated by TRPV4 activation (*Muramatsu et al., 2007*), thus decreasing hypertrophy (*Rock et al., 2008*). *SOX9*-knockin mice exhibit a dwarfism phenotype (*Amano et al., 2009*), and *SOX9* overexpression inhibits hypertrophy and endochondral ossification (*Hattori et al., 2010*; *Lui et al., 2019*), likely via parathyroid hormone-related protein (PTHrP) (*Amano et al., 2009*; *Nishimura et al., 2012b*). However, PTHrP was not strongly regulated in our data set. Furthermore, our RT-qPCR revealed that only V602I significantly up-regulated *SOX9*, and the RNAseq data showed that *SOX9* had a smaller fold change compared to other chondrogenic genes, such as *GDF5*, *COL6A1*, *COL6A3*, and *COMP*. In fact, these genes, which were up-regulated in V620I- and T89I-hiPSC-derived chondrocytes, have a pro-chondrogenic but anti-hypertrophic phenotype (*Caron et al., 2020*; *Chu et al., 2017*; *Hecht and Sage, 2006*). Therefore, these results suggest additional and alternative pathways to *FST* and *SOX9* that are responsible for the V620I and T89I skeletal dysplasias.

Our results are generally consistent with previous reports on the effects of other TRPV4 mutations such as lethal and non-lethal metatropic dysplasia-causing I604M (*Saitta et al., 2014*) and L619F (*Nonaka et al., 2019*). The data also reveal potential differences in the effects of these varying TRPV4 mutations on cell electrophysiology or differentiation. For example, we saw an increase in *SOX9* expression in V620I, while no change in T89I. Gain-of-function mutation L619F also increased *SOX9* expression (*Nonaka et al., 2019*), while I604M, which has been reported to not alter conductivity like T89I (*Kang, 2012*), decreased *SOX9* expression (*Saitta et al., 2014*). I604M also decreased *COL2A1*, *COL10A1*, and *RUNX2* expression consistent with our T89I results (*Saitta et al., 2014*). Intriguingly, the L619F mutation was reported to increase $Ca^{2+}$ signaling with activation via a TRPV4 agonist (*Nonaka et al., 2019*). However, we observed that V620I and T89I had significantly reduced $Ca^{2+}$ signaling compared to WT in response to chemical agonist GSK101, as confirmed by both confocal imaging and patch clamping. These results highlight that TRPV4 mutations have heterogeneous effects on downstream signaling pathways and thus lead to diverse disease phenotypes, despite similar classification of these mutations as 'gain-of-function'. It is also important to note that in previous studies, chondrogenic differentiation of iPSCs (*Saitta et al., 2014*) or dental pulp cells (*Nonaka et al., 2019*) were performed in short-term micromass culture, and not long-term pellet culture as in our study, potentially leading to different levels of chondrogenesis and maturation of the cells.

Our transcriptomic analysis showed significant changes in various *HOX* family genes due to TRPV4 mutations, suggesting a potential role of these genes in maintaining the immature, chondrogenic phenotype in the mutated lines. At both days 28 and 56, the top 25 up-regulated genes in the V620I and T89I lines included genes from the anterior *HOX* family (*Iimura and Pourquié, 2007*; *Seifert et al., 2015*). The high expression of anterior *HOX* genes indicates that the mutants are maintaining the chondrocytes in an early developmental stage with axial patterning. At days 28 and 56, *HOXA2*, *HOXA3*, and *HOXA4* were in the top up-regulated genes, with *HOXA4* having the largest fold change. Interestingly, gain-of-function mutations or overexpression of *HOXA2*, *HOXA3*, and *HOXA4* impair chondrogenesis, limit skeletal development, decrease endochondral ossification regulators, and delay mineralization in animal models (*Creuzet et al., 2002*; *Deprez et al., 2013*; *Kanzler et al., 1998*; *Li and Cao, 2006*; *Massip et al., 2007*; *Seifert et al., 2015*). *HOXA5* was also highly up-regulated at both days 28 and 56, and mutations in this gene showed disordered patterning of limb bud development (*Pineault and Wellik, 2014*). Finally, the rib and spine phenotypes associated with brachyolmia and metatropic dysplasia could be contributed to the altered expression of *HOXA4* to *HOXA7* as it has been shown that these genes are associated with rib and spine patterning, and alterations in expression have led to defects (*Chen et al., 1998*; *Wellik, 2009*). The only up-regulated posterior

*HOX* genes were *HOXC8* and *HOXD8* at day 56 (*Iimura and Pourquié, 2007*; *Seifert et al., 2015*). The absence of posterior *HOX9*, *HOX11*, and *HOX13*, which are associated with limb development and hypertrophic *RUNX2/3* expression (*Pineault and Wellik, 2014*; *Qu et al., 2020*), may be at least partially responsible for the improper development in skeletal dysplasias. Interestingly, many links have been identified between *HOX* genes and TGFβ3-family signaling, specifically through SMAD proteins, both within skeletal development and other processes (e.g., murine lung development) (*Li and Cao, 2006*; *Li and Cao, 2003*; *Volpe et al., 2013*).

In fact, TRPV4 and TGF-β signaling have recently been shown to interact, with effects specific to the order in which they occur (*Nims et al., 2021*; *O'Conor et al., 2014*; *Woods et al., 2021*). Consistent with previous finding with hiPSCs housing the I604M TRPV4 mutations (*Saitta et al., 2014*), the altered TRPV4 activity in our hiPSC-derived chondrocytes could be altering their response to the TGFβ3 and BMP4 treatments. Furthermore, the V620I and T89I mutations increased expression of *HTRA1*, which has been shown to bind to and alter the response to members of the TGFβ family (*Polur et al., 2010*). Furthermore, *TGFβ3* and *TWIST*, which is downstream of TGFβ3 signaling, were both up-regulated in TRPV4-mutated hiPSC-derived chondrocytes. It has been reported that *TGFB3* expression and signaling prevent osteoblastogenesis of mesenchymal stem cells (*Nishimura et al., 2012a*; *Nishimura et al., 2012b*), while *TWIST* inhibits hypertrophy regulators *RUNX2* and *FGFR2* (*Michigami, 2014*; *Miraoui and Marie, 2010*). Therefore, another mechanism of hypertrophic dysregulation with these mutations could be altered response to TGFβ family signaling.

In fact, in response to treatment with BMP4, a member of the TGFβ family, there was increased expression of *GSTA1*, which produces the antioxidant glutathione (*Chen et al., 2008*; *Hayes et al., 2005*), in WT but not in mutants. BMP4 treatment of T89I-mutated chondrocytes significantly increased expression of another antioxidant catalase (*CAT*) compared to its TGFβ3 control group; however, TRPV4-mutated chondrocytes without BMP4 treatment had significantly lower *CAT* expression. This may potentially indicate an association between antioxidants, which remove reactive oxygen species (ROS; e.g., $H_2O_2$), and chondrocyte maturation. While one study observed that chondrocyte maturation is associated with decreasing catalase (*Morita et al., 2007*), this is inconsistent with other findings. Another report stated hypoxia, which increases ROS, inhibits hypertrophic differentiation and endochondral ossification (*Leijten et al., 2012*). Many others found that ROS prevent endochondral ossification, potentially via inhibition of the hedgehog pathways (*Atashi et al., 2015*; *Chen et al., 2008*; *Fragonas et al., 1998*). Interestingly, *IHH* also had the lowest expression level in our T89I mutant chondrocytes. These findings suggest that decreased expression of *CAT* and *GSTA1* in TRPV4 mutants may also be involved in dysregulating endochondral ossification in these cells.

*GSTA1* is one of many genes with significantly lower expression in mutated chondrocytes compared to WT in response to BMP4 treatment including *ALPL*, *AMELX*, *IFITM5*, *IBSP*, and *MEPE*. These genes play important roles in bone development. For example, *IBSP* is downstream of *RUNX2*, a primary transcription factor of hypertrophic differentiation and osteoblast differentiation (*Komori, 2018*). Further, *MEPE* negatively (*Lu et al., 2004*; *Staines et al., 2012*) and *IFITM5* (*Hanagata et al., 2011*; *Moffatt et al., 2008*) and *ALPL* (*Millán, 2013*; *Strzelecka- Kiliszek et al., 2018*) positively regulate bone mineralization during skeletogenesis, respectively. Mutations in these genes also lead to bone mineralization diseases such as rickets (*Lu et al., 2004*; *Staines et al., 2012*), osteogenesis imperfecta (*Hanagata, 2016*), and hypophosphatasia with deformed long bones (*Taillandier et al., 2015*). Not only did we observe significantly lower gene expression of *ALPL* in TRPV4-mutated chondrocytes treated with BMP4; we also demonstrated that ALPL protein production is negatively associated with disease severity. Our results indicate that not only do the mutated cells have an altered hypertrophic response to BMP4, but there is a connection between these genes, particularly *ALPL*, or tissue-nonspecific alkaline phosphatase, and delayed endochondral ossification in chondrocytes bearing V620I or T89I mutations. However, how these genes and their transcription are associated with TRPV4 function and mutations still warrants further investigation.

The gene expression and protein production of *ALPL*, as well as *IHH*, in response to BMP4 treatment was not only significantly lower in mutants compared to WT, but it was also significantly lower for the severe T89I mutation compared to the moderate V620I mutation. This is one of the many targets showing the different levels of genetic and protein production between the two mutations that may be responsible for the distinct severity and phenotype of the corresponding diseases. Furthermore, several other genes were also detected to be uniquely up- or down-regulated compared to WT in

one mutated line but not the other. Interestingly, there were more uniquely expressed genes between the two mutated lines at day 56, suggesting the separation of transcriptomic profiles in V620I and T89I occurs at later time points of chondrocyte maturity. Among those unique to severe T89I include down-regulation of *IBSP*, a positive mineralization regulator, at day 28, and up-regulation of negative regulator *MEPE* at day 56. These unique genes were also not associated with many of the same biological processes as WT and V620I, especially those regarding endochondral ossification, when treated with BMP4. This, in conjunction with the high number of unique DEGs, represents a potential inhibition of hypertrophy, particularly in response to BMP4 treatment, with the T89I mutation leading to severe metatropic dysplasia. V620I also had unique differences from both WT and T89I. The decrease in mechanical properties with increased basal current of the V620I mutant was unexpected since TRPV4 activation was previously shown to increase matrix production and properties (*O'Conor et al., 2014*). Furthermore, genes uniquely up-regulated in V620I were associated with interferon type I (IFNβ). IFNβ has been reported to decrease inflammatory markers and matrix degradation (*Hu et al., 2005*; *Palmer et al., 2004*; *van Holten et al., 2004*; *Zhao et al., 2014*), despite the decrease in moduli observed in the day-42 V620I chondrogenic pellets. Interestingly, a study comparing bone marrow-derived MSCs from healthy and systemic lupus erythematous patients found that IFNβ-inhibited osteogenesis via suppression of *RUNX2* and other osteogenic genes (*Gao et al., 2020*). Highlighting a potential, unique regulator of the delayed hypertrophy in V620I leading to brachyolmia.

Here, we present multiple putative genes and pathways that could be involved in delaying, and potentially inhibiting, chondrocyte hypertrophy in V620I- and T89I-TRPV4 mutants. It should be noted, however, that this study has some potential limitations. It is well-recognized that Wnt/β-catenin signaling plays an important role in chondrocyte hypertrophy (*Hou et al., 2019*; *Huang et al., 2018*; *Michigami, 2014*). After 56 days of differentiation, we observed increased expression of β-catenin-coding gene *CTNNB1* in T89I-mutated chondrocytes highlighting this pathway could be playing a role in the inhibited hypertrophy. However, we may be preventing some hypertrophy since our chondrogenic protocol uses a pan-Wnt inhibitor to prevent off-target differentiation and promote a homogenous chondrocyte population (*Wu et al., 2021*). Nevertheless, our WT chondrocytes, but not TRPV4 mutants, exhibited hypertrophic differentiation with BMP4 treatment, suggesting that DEGs/pathways detected in our sequencing analysis are still robust. Since this study focuses on TRPV4 gain-of-function mutations, future studies could fully or partially inhibit TRPV4 signaling to determine if that would increase similarity between the mutant and WT lines at various stages of chondrogenic and hypertrophic differentiation. Additionally, this study only activated TRPV4 using the pharmacological activator GSK101. Other future experiments could activate the channel osmotically or with mechanical loading to investigate additional differences in TRPV4 function leading to skeletal dysplasias during development.

In summary, our study found that dysregulated skeletal development in the V620I- and T89I-TRPV4 dysplasias is likely due, at least in part, to delayed and inhibited chondrocyte hypertrophy. The gain-of-function mutations may lead to increased *HOX* gene expression, altered TGFβ signaling, decreased hypertrophic and biomineralization gene expression (e.g., *ALPL*, *AMELX*, *IFITM5*, *IBSP*, and *MEPE*), and genes regulating hedgehog pathways and ROS accumulation (e.g., *GSTA1* and *CAT*). Our findings lay a foundation for the development of therapeutics for these diseases and provide significant insights into the regulation of endochondral ossification via TRPV4.

## Materials and methods
### hiPSC culture

The BJFF.6 (BJFF) human iPSC line (Washington University Genome Engineering and iPSC Center [GEiC], St. Louis, MO), was used in this study as the isogenic WT control. CRISPR-Cas9 gene editing was used to create the V620I and T89I mutations in the BJFF cell line as described previously (*Adkar et al., 2019*). All three lines underwent STR profiling for cell line authentication and were verified to have no cross-contamination with other cell lines. All cells tested negative for mycoplasma. The hiPSCs were maintained on vitronectin (VTN-N; cat. num. A14700; Thermo Fisher Scientific, Waltham, MA)-coated plates in Essential 8 Flex medium (E8; cat. num. A2858501; Gibco, Thermo Fisher Scientific, Waltham, MA). Medium was changed daily until cells were passaged at 80–90% confluency (medium

supplemented with Y-27632 [cat. num. 72304; STEMCELL Technologies, Vancouver, Canada] for 24 hr) or induced into mesodermal differentiation at 30–40% confluency.

## Mesodermal differentiation

The hiPSCs were differentiated through the mesodermal pathway as previously described (*Adkar et al., 2019*; *Dicks et al., 2020*; *Wu et al., 2021*). In brief, cells were fed daily with different cocktails of growth factors and small molecules for 12 days in mesodermal differentiation medium and driven through the anterior primitive streak (1 day; 30 ng/ml Activin [cat. num. 338-AC; R&D Systems, Minneapolis, MN], 20 ng/ml FGF2 [cat. num. 233-FB-025/CF; R&D Systems, Minneapolis, MN], 4 μM CHIR99021 [cat. num. 04-0004-02; Reprocell, Beltsville, MD]), paraxial mesoderm (1 day; 20 ng/ml FGF2, 3 μM CHIR99021, 2 μM SB505124 [cat. num. 3263; Tocris Bioscience, Bristol, UK], 4 μM dorsomorphin [DM; cat. num. 04-0024; Reprocell, Beltsville, MD]), early somite (1 day; 2 μM SB505124, 4 μM dorsomorphin, 500 nM PD173074 [cat. num. 3044; Tocris Bioscience, Bristol, UK], 1 μM Wnt-C59 [cat. num. C7641-2s; Cellagen Technologies, San Diego, CA]), and sclerotome (3 days; 1 μM Wnt-C59, 2 μM purmorphamine [cat. num. 04-0009; Reprocell, Beltsville, MD]) into chondroprogenitor cells (6 days; 20 ng/ml BMP4 [cat. num. 314-BP-010CF; R&D Systems, Minneapolis, MN]). Mesodermal differentiation medium had a base of Iscove's Modified Dulbecco's Medium, glutaMAX (IMDM; cat. num. 31980097; Gibco, Thermo Fisher Scientific, Waltham, MA) and Ham's F-12 nutrient mix, glutaMAX (F12; cat. num. 31765092; Gibco, Thermo Fisher Scientific, Waltham, MA) in equal parts supplemented with 1% penicillin–streptomycin (P/S; cat. num. 15140122; Gibco, Thermo Fisher Scientific, Waltham, MA), 1% Insulin–Transferrin–Selenium (ITS+; cat. num. 41400045; Gibco, Thermo Fisher Scientific, Waltham, MA), 1% chemically defined concentrated lipids (cat. num. 11905031; Thermo Fisher Scientific, Waltham, MA), and 450 μM 1-thioglycerol (cat. num. M6145; Millipore Sigma, St. Louis, MO). The chondroprogenitor cells were then disassociated for chondrogenic differentiation.

## Chondrogenic differentiation with 3D pellet culture

Cells were differentiated into chondrocytes using a high-density, suspension pellet culture (*Adkar et al., 2019*; *Dicks et al., 2020*; *Wu et al., 2021*). In summary, cells were resuspended in chondrogenic medium: Dulbecco's Modified Eagle Medium/F12, glutaMAX (DMEM/F12; cat. num. 10565042; Gibco, Thermo Fisher Scientific, Waltham, MA), 1% P/S, 1% ITS+, 1% Modified Eagle Medium (MEM) with nonessential amino acids (NEAA; cat. num. 11140050; Gibco, Thermo Fisher Scientific, Waltham, MA), 0.1% dexamethasone (Dex; cat. num. D4902; Millipore Sigma, St. Louis, MO), and 0.1% 2-Mercaptoethnol (2-ME; cat. num. 21985023; Gibco, Thermo Fisher Scientific, Waltham, MA) supplemented with 0.1% L-ascorbic acid (ascorbate; cat. num. A8960; Millipore Sigma, St. Louis, MO), 0.1% L-proline (proline; cat. num. P5607; Millipore Sigma, St. Louis, MO), 10 ng/ml human transforming growth factor- β3 ( TGFβ3; cat. num. 243-B3-010/CF; R&D Systems, Minneapolis, MN), 1 μM Wnt-C59, and 1 μM ML329 (cat. num. 22481; Cayman Chemical, Ann Arbor, MI) at $5 \times 10^5$ cells/ml. One mL of the cell solution was added to a 15-ml conical tube (cat. num. 430790; Corning, Corning, NY) and centrifuged to form the spherical pellets. Pellets were fed every 3–4 days with complete chondrogenic medium until the desired time point. Several time points of the chondrogenic pellets were used to study chondrocyte maturation (7, 14, 28, and 42 days), mechanical properties (28 and 42 days), hypertrophy (28 days) or, after digestion to single-cell day-28 chondrocytes, on $Ca^{2+}$ signaling in response to pharmacological activation of TRPV4.

## BMP4 treatment to promote hypertrophic differentiation

Some day-28 pellets were also further differentiated for an additional 4 weeks to examine the effects of the mutations on chondrocyte hypertrophy. Pellets were cultured with complete chondrogenic medium with either TGFβ3 (10 ng/ml) alone, BMP4 (50 ng/ml) alone, or a combination of TGFβ3 (10 ng/ml) and BMP4 (50 ng/ml).

## Dissociation of chondrogenic pellets to obtain single-cell hiPSC-derived chondrocytes

To isolated hiPSC-derived chondrocytes, day-28 chondrogenic pellets were rinsed and placed in an equal volume (1 pellet per 1 ml) of digestion medium (0.4% wt/vol type II collagenase [cat. num. LS00417; Worthington Biochemical, Lakewood, NJ] in DMEM/F12 with 10% fetal bovine serum [FBS;

cat. num. S11550; Atlanta Biologicals, R&D Systems, Minneapolis, MN]). The tubes were placed on an orbital shaker at 37°C and vortexed every 20 min for approximately 2 hr. Once the tissue was digested and could no longer be seen by the naked eye, the digestion medium was neutralized in DMEM/F12 medium containing 10% FBS. These cells were used for patch clamping and confocal experiments.

### TRPV4 agonists and antagonists

Solutions were prepared immediately before experiments and held at room temperature. GSK1016790A (GSK101; cat. num. G0798; Sigma-Aldrich, St. Louis, MO) and/or GSK205 (cat. num. AOB1612 1263130-79-5; AOBIOUS, Gloucester, MA), in addition to dimethyl sulfoxide (DMSO) for a vehicle control, were added to assay buffer (Hanks' Balanced Salt Solution [HBSS; cat. num. 14025076; Gibco, Thermo Fisher Scientific, Waltham, MA] with 2% N-2-hydroxyethylpiperazine-N'-2-ethanesulfonic acid (HEPES) [cat. num. 15630130; Gibco, Thermo Fisher Scientific, Waltham, MA]) at 2-folds of the desired concentration (20 nM GSK101, 40 µM GSK205). Solutions were made at 2-folds of the desired concentration because they would be mixed at an equal volume of assay buffer after capturing a baseline fluorescence in $Ca^{2+}$ signaling experiments.

### Patch clamping

Isolated chondrocytes were kept on ice and used for patching within 36 hr. Patch-clamp experiments were carried out at RT under two conditions. Single-channel measurements were made in excised inside-out membrane patches in a symmetric potassium chloride (KCl) solution (148 mM KCl, 1 mM $K_2$EDTA, 1 mM ethylene glycol-bis(β-aminoethyl ether)-N,N,N',N'-tetraacetic acid (egtazic acid; EGTA), 10 mM HEPES, pH 7.4). Channel activation was achieved by bath perfusion with the same buffer solution containing 10 nM GSK101. Blocking was performed using the same buffer solution supplied with both 10 nM GSK101 and 20 µM GSK205. Recordings were made at −30 mV membrane. Whole-cell currents were recorded using an external sodium chloride (NaCl) solution (150 mM NaCl, 5 mM KCl, 1 mM EGTA, 10 mM glucose, 10 mM HEPES, and 10 µM free $Ca^{2+}$) and KCl pipette solution as used for single-channel recordings. Inhibition of basal currents was performed by pre-incubation of the cells in external solution supplied with 20 µM GSK205 for 20 min before patching; the drug was also present in the bath at the same concentration during the experiment. Data were acquired at 3 kHz, low-pass filtered at 1 kHz with Axopatch 1D patch-clamp amplifier and digitized with Digidata 1320 digitizer (Molecular Devices, San Jose, CA). Data analysis was performed using the pClamp software suite (Molecular Devices, San Jose, CA). Pipettes with 2.0–4.0 MOhm resistance in symmetric 150 mM KCl buffer were pulled from Kimble Chase 2502 soda lime glass with a Sutter P-86 puller (Sutter Instruments, Novato, CA).

### Confocal imaging of $Ca^{2+}$ signaling

hiPSC-derived chondrocytes from digested pellets were plated in DMEM medium containing 10% FBS at $2.1 \times 10^4$ cells/$cm^2$ in 35-mm dishes for 6–8 hr to allow the cells to adhere without dedifferentiating. Cells were then rinsed and stained for 30 min with Fluo-4 AM (cat. num. F14201; Thermo Fisher Scientific, Waltham, MA), Fura Red AM (cat. num. F3021; Thermo Fisher Scientific, Waltham, MA), and sulfinpyrazone (cat. num. S9509-5G; Sigma-Aldrich, St. Louis, MO) with 20 mM GSK205 or 1000× DMSO (vehicle control). The dye solution was replaced with assay buffer before imaging cells on a confocal microscope (LSM 880; Zeiss, Oberkochen, Germany) at baseline for the first 100 frames (approximately 6 min). Then, an equal volume of a 2× solution of GSK101 or GSK101 and GSK205 was added, and imaging continued for an additional 300 frames (approximately 20 min). Fiji software (ImageJ, version 2.1.0) was used to locate cells and quantify the ratiometric fluorescence intensity ($Intensity_{fluo-4}/Intensity_{fura\ red}$). In brief, .czi files were imported into Fiji and the channels were split. After applying the median filter, the image calculator divided the green channel by the red. A Z-projection was performed based on the maximum fluorescence of the red channel (to ensure that all cells were identified even in groups were there was no increase in $Ca^{2+}$ signaling). A threshold and watershed binary were then applied, and measurements were set for a cell size of 100-infinity. Outlines were projected, and the mean fluorescence of each cell was measured over time. The average fluorescence was plotted for all the cells in the group over time. Area under the curve and time of response were calculated to quantify differences between groups. Cells were classified as responders if they had a fluorescence greater than the baseline mean plus 3 times the standard deviation in at least a quarter

of the frames. Time of response was the time of the first frame in which the cell responded for at least two consecutive frames. The fluorescence was measured for all the cells in the frame of view as technical replicates for two experimental replicates.

## AFM measurement of neocartilage mechanical properties

Day-28 and -42 hiPSC-derived pellets were rinsed in phosphate-buffered saline (PBS) and snap frozen in optimal cutting temperature (cat. num. 4583; Sakura Finetek, Torrance, CA) medium and stored at −80 °C. Pellets were cryosectioned using cryofilm (type 2C(10); Section-Lab, Hiroshima, Japan) in multiple different regions of the pellet (i.e., zones). The 10 µm cryosection with cryofilm was fixed on a microscope slide using chitosan and stored at 4°C overnight. The next day, cryosections were mechanically loaded using an AFM (MFP-3D Bio, Asylum Research, Goleta, CA) as previously described (*Votava et al., 2019*). Briefly, the samples were tested in PBS at 37°C to maintain hydration and mimic physiologic conditions, respectively. The sections were mechanically probed using a silicon cantilever with a spherical tip (5 µm diameter, $k \sim 7.83$ N/m, Novascan Technologies, Ames, IA). An area of 10 µm$^2$ with 0.5 µm intervals (400 indentations) was loaded to 300 nN with the loading rate of 10 µm/s. Multiple locations from different sites of each zone and pellet were loaded as replicates. The curves obtained from AFM were imported into a custom written MATLAB code to determine the mechanical properties of the pellets. Using contact point extrapolation, the contact point between the cantilever's tip and the tissue was detected, and the elastic modulus was calculated using a modified Hertz model (*Darling et al., 2010*; *Darling et al., 2006*; *Votava et al., 2019*; *Wilusz et al., 2013*; *Zelenski et al., 2015*). This code is available at: https://github.com/guilak-lab/programs/tree/guilak-lab-TRPV4-paper (copy archived at swh:1:rev:465cfaeea5676c514c264785b5db626513baa0d1; *Dicks et al., 2022*).

## Histology

Chondrogenic pellets at days 7, 14, 28, 42, and 56 (with and without BMP4) were fixed and dehydrated in sequential steps of increasing ethanol and xylene solutions until embedded in paraffin wax. Wax blocks were cut into 8 µm sections on microscope slides for histological and immunohistochemical analysis. Slides were rehydrated in ethanol and water and the nuclei were stained with Harris hematoxylin and sGAGs with Safranin-O. Antigen retrieval was performed on rehydrated slides followed by blocking, the addition of primary and secondary antibodies, and AEC development to label collagen proteins (COL1A1, COL2A1, COL6A1, and COL10A1) and Vector Hematoxylin QS counterstain (cat. num. H-3404, Vector Laboratories, Newark, CA).

## Biochemical analysis

Chondrogenic pellets at days 7, 14, 28, and 42 were washed with PBS and digested in papain overnight at 65°C. sGAG and dsDNA content were measured using the dimethylmethylene blue (DMMB; cat. num. 341088, Sigma-Aldrich, St. Louis, MO) and PicoGreen assays (Quant-iT PicoGreen dsDNA Assay Kit; cat. num. P7589; Thermo Fisher Scientific, Waltham, MA), respectively. sGAG content was normalized to dsDNA. Three to four independent experiments were performed with 3–4 technical replicates per group.

## Western blot

Day-56 pellets treated with TGFβ3, TGFβ3 + BMP4, or BMP4 were digested to single cells, as described above, and lysed in RIPA buffer (cat. num. 9806S; Cell Signaling Technology, Danvers, MA) with protease inhibitor (cat. num. 87786; Thermo Fisher Scientific, Waltham, MA). Protein concentration was then measured using the BCA Assay (Pierce). Twenty micrograms of protein for each well were separated on 10% sodium dodecyl sulfate–polyacrylamide gel electrophoresis gel with pre-stained molecular weight markers (cat. num. 161-0374; Bio-Rad, Hercules, CA) and transferred to a polyvinylidene fluoride (PVDF) membrane. The PVDF membrane blots were incubated overnight at 4°C with the primary antibodies, respectively: anti-COL10A1 (1:500; cat. num. PA5-97603; Thermo Fisher Scientific, Waltham, MA), anti-RUNX2 (1:2000; cat. num. 41-1400, Thermo Fisher Scientific), anti-MMP13 (1;2000; cat. num. MA5-14238; Thermo Fisher Scientific, Waltham, MA), anti-IHH (1:500; cat. num. MA5-37541; Thermo Fisher Scientific, Waltham, MA), anti-ALPL (1:3000; cat. num. MAB29092, R&D systems), and anti-GAPDH (1:30000; cat. num. 60004-1-Ig; Proteintech, Rosemont, IL) as the loading control. TidyBlot-Reagent-HRP (1:1000; cat. num. 147711; Bio-Rad, Hercules, CA)

and horse anti-mouse IgG secondary antibody (1:3000; cat. num. 7076; Cell Signaling, Danvers, MA) were used accordingly. Immunoblots were imaged using the iBright FL1000 Imaging System (Thermo Fisher Scientific, Waltham, MA). Using photoshop, the images were inverted, and the protein abundance of each band was quantified by multiplying the mean of signal intensity by the pixels of the individual band. The relative protein abundance was normalized to the GAPDH levels. The maximum value was arbitrarily set to 1.

## RNA isolation

Chondrogenic pellets at days 7, 14, 28, 42, and 56 were washed with PBS, lysed, snap frozen, and homogenized. RNA was isolated using the Total RNA Purification Plus Kit (cat. num. 48400; Norgen Biotek, Thorold, Canada) and used immediately for either RT-qPCR or RNA-seq.

## Gene expression with RT-qPCR

Isolated RNA was reverse transcribed into cDNA. The cDNA was used to run real-time, quantitative PCR using Fast SYBR Green Master Mix (cat. num. 4385610; Thermo Fisher Scientific, Waltham, MA). Gene expression was analyzed using the $\Delta\Delta C_T$ method with hiPSC as the reference time point and *TBP* as the housekeeping gene (*Livak and Schmittgen, 2001*). Three to four independent experiments were performed with 3–4 technical replicates per group. Primers can be found in *Figure 3—figure supplement 1*.

## Genome-wide mRNA sequencing

Isolated RNA was treated with DNase (cat. num. 25720; Norgen Biotek, Thorold, Canada) and cleaned (cat. num. 43200; Norgen Biotek, Thorold, Canada) according to the manufacturer's instructions prior to submitting to the Genome Technology Access Center at Washington University in St. Louis (GTAC). Libraries were prepared according to the manufacturer's protocol. Samples were indexed, pooled, and sequenced at a depth of 30 million reads per sample on an Illumina NovaSeq 6000. Basecalls and demultiplexing were performed with Illumina's bcl2fastq software and a custom python demultiplexing program with a maximum of one mismatch in the indexing read. RNA-seq reads were then aligned to the Ensembl release 76 primary assembly with STAR version 2.5.1a (*Dobin et al., 2013*). Gene counts were derived from the number of uniquely aligned unambiguous reads by Subread:featureCount version 1.4.6-p5 (*Liao et al., 2014*). Isoform expression of known Ensembl transcripts were estimated with Salmon version 0.8.2 (*Patro et al., 2017*). Sequencing performance was assessed for the total number of aligned reads, total number of uniquely aligned reads, and features detected. The ribosomal fraction, known junction saturation, and read distribution over known gene models were quantified with RSeQC version 2.6.2 (*Wang et al., 2012*).

## Transcriptomic analysis of sequencing datasets

R and the DESeq2 package were used to read un-normalized gene counts, and genes were removed if they had counts lower than 200 (*Love et al., 2014*). Regularized-logarithm transformed data of the samples were visualized with the *Pheatmap* package (*Kolde, 2015*) function on the calculated Euclidean distances between samples or with the *ggplot2* package (*Wickham, 2009*) to create a PCA. The transformed data were also used to determine the top 5000 most variable genes across the samples. The replicates, from DESeq data, for each group were averaged together, and the up- and down-regulated DEGs were determined. The total number of DEGs was plotted using GraphPad Prism. At day 28, the V620I and T89I lines were compared to WT. At day 56, TGFβ3-treated V620I and T89I were compared to TGFβ3-treated WT, and BMP4-treated groups were compared to their respective TGFβ3-treated group of the same line (e.g., BMP4-treated WT vs. TGFβ3-treated WT). Genes were considered differentially expressed if adjusted p value ($p_{adj}$) <0.1 and log$_2$(fold change) ≥1 or ≤−1. The intersecting and unique DEGs were determined and plotted with the *intersect* and *setdiff*, and *venn.diagram* functions (*VennDiagram* package; *Chen and Boutros, 2011*). The fold changes of common chondrogenic, hypertrophic, growth factor, Ca$^{2+}$ signaling, and off-target genes, in the top 5000 most variable genes, were plotted using the *pheatmap* function. The top 25 most up- and down-regulated for each group, based on log$_2$(fold change), and the log$_2$(fold change) of that gene for the other group(s) were also plotted with the *pheatmap*. Gene lists (e.g., intersected genes, genes up-regulated with BMP4 treatment) were entered into g:profiler to determine associated GO

Biological Processes, Molecular Functions, Cellular Components, KEGG pathways, Reactome pathways, and Human Phenotype (HP) Ontologies (*Raudvere et al., 2019*). The negative $\log_{10}$ of the adjusted p value for each term was plotted with GraphPad Prism or using a function to scale circle diameter to the p value in Illustrator.

The gap statistic method determined the ideal number of clusters resulting from BMP4 treatment was either 1 or 9. We then performed *k*-means clustering with 9 clusters and plotted the gene expression trends for each gene within the cluster with the average expression trend overlaying for each cell line of the largest cluster using the *tidyverse* package (*Altman and Krzywinski, 2017*). The genes in each cluster, with the normalized counts for each group, are listed in *Supplementary file 1*. The largest cluster was plotted using the Cytoscape String app's protein interaction to create a protein–protein network (*Doncheva et al., 2019*; *Shannon et al., 2003*). Using the average log fold change with BMP4 treatment across lines, the network was propagated using the Diffusion app, and functional enrichment with EnrichmentMap was performed on the network (*Merico et al., 2010*). We then created a network connecting the genes to their associated genes with black lines and to their associated GO processes using gray lines. We colored the gene circles with three colors representing the log fold change of that gene in each line. The white arrows were added to the color scale legend to indicate maximum log fold change for each line.

## Statistical analysis

Data were graphed and analyzed using GraphPad Prism (Version 9.1.0). Outliers were removed from the data using the ROUT method ($Q = 1\%$), and the data were tested for normality with the Shapiro–Wilk test ($a = 0.05$). For RT-qPCR, normally distributed data were analyzed within each time point using a Brown–Forsythe and Welch one-way analysis of variance (ANOVA) with multiple comparisons (mean of each column, cell line, with every other column). A Kruskal–Wallis test was used if data were not normally distributed. For biochemical analysis, mechanical properties, and area under the curve, and time of response, data were analyzed using an ordinary two-way ANOVA, comparing each cell with all other cells, with Tukey's post hoc test. Area under the curve was quantified for plots over time considering a baseline of $Y = 0$, ignoring peaks less than 10% of the distance from minimum to maximum $Y$, and all peaks going over the baseline.

## Acknowledgements

This work was supported by Shriners Hospitals for Children – St. Louis, the National Institutes of Health (R01 AG46927, R01 AG15768, R01 AR072999, R00 AR075899, P30 AR073752, P30 AR074992, T32 DK108742, T32 EB018266, and CTSA grant UL1 TR002345). We would like to thank the Washington University Genome Engineering and iPSC Center and the Genome Technology Access Center (GTAC) for their assistance with the CRISPR-Cas9 editing and RNA sequencing, respectively. We would also like to thank Dr. Monica Sala-Rabanal for assistance and advice in the initial aspects of this study.

## Additional information

### Competing interests

Wolfgang Liedtke: Patents on TRPV4 inhibitors have been licensed to TRPblue (US Patents 9,701,675; 10,329,265; and 11,014,896). Dr. Liedtke is an executive employee of Regeneron Pharmaceuticals (Tarrytown NY).. Farshid Guilak: Patents on TRPV4 inhibitors licensed to TRPblue Inc (US Patents 9,701,675; 10,329,265; and 11,014,896). The other authors declare that no competing interests exist.

### Funding

| Funder | Grant reference number | Author |
|---|---|---|
| National Institutes of Health | AG15768 | Farshid Guilak |
| National Institutes of Health | AG46927 | Farshid Guilak |

| Funder | Grant reference number | Author |
|--------|------------------------|--------|
| National Institutes of Health | ar072999 | Farshid Guilak |
| National Institutes of Health | AR075899 | Chia-Lung Wu |

The funders had no role in study design, data collection, and interpretation, or the decision to submit the work for publication.

## Author contributions

Amanda R Dicks, Conceptualization, Data curation, Formal analysis, Validation, Writing – original draft; Grigory I Maksaev, Zainab Harissa, Data curation, Formal analysis, Investigation, Writing – review and editing; Alireza Savadipour, Data curation, Formal analysis, Investigation, Visualization, Methodology, Writing – review and editing; Ruhang Tang, Data curation, Formal analysis, Investigation, Visualization, Writing – review and editing; Nancy Steward, Conceptualization, Data curation, Formal analysis, Supervision, Funding acquisition, Investigation, Writing – review and editing; Wolfgang Liedtke, Conceptualization, Supervision, Funding acquisition, Writing – review and editing; Colin G Nichols, Conceptualization, Supervision, Investigation, Writing – review and editing; Chia-Lung Wu, Conceptualization, Resources, Supervision, Funding acquisition, Investigation, Writing – review and editing; Farshid Guilak, Conceptualization, Resources, Data curation, Formal analysis, Supervision, Funding acquisition, Investigation, Writing – review and editing

## Author ORCIDs

Amanda R Dicks ⓘ http://orcid.org/0000-0002-6036-280X
Colin G Nichols ⓘ http://orcid.org/0000-0002-4929-2134
Chia-Lung Wu ⓘ http://orcid.org/0000-0001-9598-7036
Farshid Guilak ⓘ http://orcid.org/0000-0001-7380-0330

## Decision letter and Author response

Decision letter https://doi.org/10.7554/eLife.71154.sa1
Author response https://doi.org/10.7554/eLife.71154.sa2

# Additional files

## Supplementary files

• Supplementary file 1. Additional figures to support data in *Figures 2, 3 and 5*. *Figure 2—figure supplement 1*. Matrix production and mechanical properties through day 42 of chondrogenic differentiation. *Figure 3—figure supplement 1*. Gene expression using RT-qPCR through day 42 of chondrogenic differentiation. *Figure 4—figure supplement 1*. Top 15 mutant-specific differentially expressed genes (DEGs) compared to wildtype (WT) and genes of interest at days 28 and 56 as identified by RNA sequencing. *Figure 4—figure supplement 2*. Top 25 most up- and down-regulated DEGs compared to WT at day 56 as identified by RNA sequencing. *Figure 5—figure supplement 1*. Hypertrophic gene and protein expression in TGFβ3- and BMP4-treated day-56 chondrogenic pellets.

• Transparent reporting form

## Data availability

All RNAseq data files generated and reported in this study are available on GEO (accession number GSE225446, https://www.ncbi.nlm.nih.gov/geo/query/acc.cgi?acc=GSE225446).

The following dataset was generated:

| Author(s) | Year | Dataset title | Dataset URL | Database and Identifier |
|-----------|------|---------------|-------------|-------------------------|
| Dicks AR, Maksaev GI, Harissa Z, Savadipour A, Tang R, Steward N, Liedtke W, Nichols CG, Wu C, Guilak F | 2023 | Skeletal dysplasia-causing TRPV4 mutations suppress the hypertrophic differentiation of human iPSC-derived chondrocytes | https://www.ncbi.nlm.nih.gov/geo/query/acc.cgi?acc=GSE225446 | NCBI Gene Expression Omnibus, GSE225446 |

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

## Appendix 1

### Appendix 1—key resources table

| Reagent type (species) or resource | Designation | Source or reference | Identifiers | Additional information |
|---|---|---|---|---|
| Gene (*Homo sapien*) | *TPRV4*; Transient Receptor Potential Cation Channel Subfamily V Member 4 | HGNC Symbol | HGNC:18083; ENSEMBL:ENSG00000111199 | |
| Gene (*Homo sapien*) | *SOX9*; SRY-box transcription factor 9 | HGNC Symbol | HGNC:11204; ENSEMBL:ENSG00000125398 | |
| Gene (*Homo sapien*) | *RUNX2*; RUNX family transcription factor 2 | HGNC Symbol | HGNC:10472; ENSEMBL:ENSG00000124813 | |
| Gene (*Homo sapien*) | *FST*; follistatin | HGNC Symbol | HGNC:3971; ENSEMBL:ENSG00000134363 | |
| Gene (*Homo sapien*) | *ACAN*; aggrecan | HGNC Symbol | HGNC:319; ENSEMBL:ENSG00000157766 | |
| Gene (*Homo sapien*) | *COL2A1*; collagen type II alpha 1 chain | HGNC Symbol | HGNC:2200; ENSEMBL:ENSG00000139219 | |
| Gene (*Homo sapien*) | *S100B*; S100 calcium-binding protein B | HGNC Symbol | HGNC:10500; ENSEMBL:ENSG00000160307 | |
| Gene (*Homo sapien*) | *COL1A1*; collagen type I alpha 1 chain | HGNC Symbol | HGNC:2197; ENSEMBL:ENSG00000108821 | |
| Gene (*Homo sapien*) | *COL10A1*; collagen type X alpha 1 chain | HGNC Symbol | HGNC:2185; ENSEMBL:ENSG00000123500 | |
| Gene (*Homo sapien*) | *ALPL*; alkaline phosphatase, biomineralization associated | HGNC Symbol | HGNC:438; ENSEMBL:ENSG00000162551 | |
| Gene (*Homo sapien*) | *IHH*; Indian hedgehog signaling molecule | HGNC Symbol | HGNC:5956; ENSEMBL:ENSG00000163501 | |
| Gene (*Homo sapien*) | *GSTA1*; glutathione *S*-transferase alpha 1 | HGNC Symbol | HGNC:4626; ENSEMBL:ENSG00000243955 | |
| Gene (*Homo sapien*) | *AMELX*; amelogenin X-linked | HGNC Symbol | HGNC:461; ENSEMBL:ENSG00000125363 | |
| Gene (*Homo sapien*) | *IFITM5*; interferon induced transmembrane protein 5 | HGNC Symbol | HGNC:16644; ENSEMBL:ENSG00000206013 | |
| Gene (*Homo sapien*) | *IBSP*; integrin-binding sialoprotein | HGNC Symbol | HGNC:5341; ENSEMBL:ENSG00000029559 | |
| Gene (*Homo sapien*) | *MEPE*; matrix extracellular phosphoglycoprotein | HGNC Symbol | HGNC:13361; ENSEMBL:ENSG00000152595 | |
| Cell line (*Homo sapien*) | BJFF.6; BJFF | Washington University Genome Engineering and iPSC Center | RRID:CVCL_VU02 | Induced pluripotent stem cell derived from foreskin fibroblast |
| Cell line (*Homo sapien*) | V620I | This paper | | Washington University Genome Engineering and iPSC Center; CRISPR-edited BJFF.6 with V620I TRPV4 mutation |
| Cell line (*Homo sapien*) | T89I | This paper | | Washington University Genome Engineering and iPSC Center; CRISPR-edited BJFF.6 with T89I TRPV4 mutation |
| Antibody | Human Alkaline Phosphatase/ ALPL Antibody; Anti-ALPL (mouse monoclonal) | R&D Systems | Cat #: MAB29092; RRID:AB_2924405 | WB (1:3000) |
| Antibody | Anti-Collagen I antibody; Anti-COL1A1 (mouse monoclonal) | Abcam | Cat #: ab90395; RRID:AB_2049527 | IHC P (1:800); pepsin retrieval (5 min, RT) |
| Antibody | Collagen type II: Anti-COL2A1 (mouse monoclonal) | Iowa Hybridoma Bank | Cat #: II-II6B3-s; RRID:AB_528165 | IHC P (1:10); proteinase k retrieval (3 min, 37°C) |
| Antibody | Collagen Type VI antibody; Anti-COL6A1 (rabbit polyclonal) | Fitzgerald Industries | Cat #: 70F-CR009X; RRID:AB_1283876 | IHC P (1:1000); proteinase k retrieval (3 min, 37°C) |

*Appendix 1 Continued on next page*

*Appendix 1 Continued*

| Reagent type (species) or resource | Designation | Source or reference | Identifiers | Additional information |
|---|---|---|---|---|
| Antibody | Monoclonal Anti-Collagen, Type X antibody produced in mouse; Anti-COL10A1 (mouse monoclonal) | Millipore Sigma | Cat #: C7974; RRID:AB_259075 | IHC P (1:200); pepsin retrieval (5 min, RT) |
| Antibody | Collagen X Polyclonal Antibody; anti-COL10A1 (rabbit polyclonal) | Thermo Fisher Scientific | Cat #: PA5-97603; RRID:AB_2812218 | WB (1:500) |
| Antibody | GAPDH Monoclonal antibody; anti-GAPDH (mouse monoclonal) | Proteintech | Cat #: 60004-1-Ig; RRID:AB_2107436 | WB (1:30,000) |
| Antibody | IHH Monoclonal Antibody (363CT4.1.6); Anti-IHH (mouse monoclonal) | Thermo Fisher Scientific | Cat #: MA5-37541; RRID:AB_2897471 | WB (1:500) |
| Antibody | MMP13 Monoclonal Antibody (VIIIA2); Anti-MMP13 (mouse monoclonal) | Thermo Fisher Scientific | Cat #: MA5-14238; RRID:AB_10981616 | WB (1:2000) |
| Antibody | RUNX2 Monoclonal Antibody (ZR002); Anti-RUNX2 (mouse monoclonal) | Thermo Fisher Scientific | Cat #: 41-1400 RRID: AB_2533497 | WB (1:2000) |
| Antibody | Anti-mouse IgG, HRP-linked antibody; horse anti-mouse IgG secondary antibody (horse polyclonal) | Cell Signaling | Cat #: 7076; RRID:AB_330924 | WB (1:30,000) |
| Antibody | Goat Anti-Mouse IgG H&L (Biotin); Goat anti-mouse antibody (goat polyclonal) | Abcam | Cat #: ab97021; RRID:AB_10679674 | IHC (1:500) |
| Antibody | Goat Anti-Rabbit IgG H&L (Biotin); Goat anti-rabbit antibody (goat polyclonal) | Abcam | Cat #: ab6720; RRID:AB_954902 | IHC (1:500) |
| Sequence-based reagent | ACAN_F | *Huynh et al., 2020* | PCR primers | CACTTCTGAGTTCGTGGAGG |
| Sequence-based reagent | ACAN_R | *Huynh et al., 2020* | PCR primers | ACTGGACTCAAAAAGCTGGG |
| Sequence-based reagent | COL1A1_F | *Adkar et al., 2019* | PCR primers | TGTTCAGCTTTGTGGACCTC |
| Sequence-based reagent | COL1A1_R | *Adkar et al., 2019* | PCR primers | TTCTGTACGCAGGTGATTGG |
| Sequence-based reagent | COL2A1_F | *Adkar et al., 2019* | PCR primers | GGCAATAGCAGGTTCACGTA |
| Sequence-based reagent | COL2A1_R | *Adkar et al., 2019* | PCR primers | CTCGATAACAGTCTTGCCCC |
| Sequence-based reagent | COL10A1_F | *Adkar et al., 2019* | PCR primers | CATAAAAGGCCCACTACCCAAC |
| Sequence-based reagent | COL10A1_R | *Adkar et al., 2019* | PCR primers | ACCTTGCTCTCCTCTTACTGC |
| Sequence-based reagent | FST_F | *Ohta et al., 2015* | PCR primers | TGTGCCCTGACAGTAAGTCG |
| Sequence-based reagent | FST_R | *Ohta et al., 2015* | PCR primers | GTCTTCCGAAATGGAGTTGC |
| Sequence-based reagent | S100B_F | *Dix et al., 2016* | PCR primers | AGGGAGGGAGACAAGCACAA |
| Sequence-based reagent | S100B_R | *Dix et al., 2016* | PCR primers | ACTCGTGGCAGGCAGTAGTA |
| Sequence-based reagent | SOX9_F | *Loh et al., 2016* | PCR primers | CGTCAACGGCTCCAGCAAGAACAA |
| Sequence-based reagent | SOX9_R | *Loh et al., 2016* | PCR primers | GCCGCTTCTCGCTCTCGTTCAGAAGT |
| Sequence-based reagent | TRPV4_F | *Luo et al., 2018* | PCR primers | AGAACTTGGGCATCATCAACGAG |

*Appendix 1 Continued on next page*

*Appendix 1 Continued*

| Reagent type (species) or resource | Designation | Source or reference | Identifiers | Additional information |
|---|---|---|---|---|
| Sequence-based reagent | TRPV4_R | *Luo et al., 2018* | PCR primers | GTTCGAGTTCTTGTTCAGTTCCAC |
| Sequence-based reagent | TBP_F | *Adkar et al., 2019* | PCR primers | AACCACGGCACTGATTTTCA |
| Sequence-based reagent | TBP_R | *Adkar et al., 2019* | PCR primers | ACAGCTCCCCACCATATTCT |
| Peptide, recombinant protein | Vitronectin; VTN-N | Thermo Fisher Scientific | Cat #: A14700 | |
| Peptide, recombinant protein | Activin | R&D Systems | Cat #: 338-AC | |
| Peptide, recombinant protein | Fibroblastic growth factor 2; FGF2 | R&D Systems | Cat #: 233-FB-025/CF | |
| Peptide, recombinant protein | Bone morphogenetic protein 4; BMP4 | R&D Systems | Cat #: 314-BP-010CF | |
| Peptide, recombinant protein | Human transforming growth factor-$\beta$3; TGF$\beta$3 | R&D Systems | Cat #: 243-B3-010/CF | |
| Peptide, recombinant protein | Type II collagenase | Worthington Biochemical | Cat #: LS00417 | Activity 225 u/ML |
| Commercial assay or kit | Fluo-4 AM | Thermo Fisher Scientific | Cat #: F14201 | |
| Commercial assay or kit | Fura Red AM | Thermo Fisher Scientific | Cat #: F3021 | |
| Commercial assay or kit | Quant-iT PicoGreen dsDNA Assay Kit; PicoGreen | Thermo Fisher Scientific | Cat #: P7589 | |
| Commercial assay or kit | Total RNA Purification Plus Kit | Norgen Biotek | Cat #: 48400 | |
| Commercial assay or kit | Fast SYBR green | Thermo Fisher Scientific | Cat #: 4385610 | |
| Commercial assay or kit | Histostain Plus Kit | Thermo Fisher Scientific | Cat #: 858943 | |
| Commercial assay or kit | AEC substrate solution | Abcam | Cat #: ab64252 | |
| Chemical compound, drug | Y-27632 | STEMCELL Technologies | Cat #: 72304 | |
| Chemical compound, drug | ReLeSR | STEMCELL Technologies | Cat #: 053263872 | |
| Chemical compound, drug | CHIR99021 | Reprocell | Cat #: 04-0004-02 | |
| Chemical compound, drug | SB505124 | Tocris Bioscience | Cat #: 3263 | |
| Chemical compound, drug | Dorsomorphin; DM | Reprocell | Cat #: 04-0024 | |
| Chemical compound, drug | PD173074 | Tocris Bioscience | Cat #: 3044 | |
| Chemical compound, drug | Wnt-C59 | Cellagen Technologies | Cat #: C7641-2s | |

*Appendix 1 Continued on next page*

*Appendix 1 Continued*

| Reagent type (species) or resource | Designation | Source or reference | Identifiers | Additional information |
|---|---|---|---|---|
| Chemical compound, drug | Purmorphamine | Reprocell | Cat #: 04-0009 | |
| Chemical compound, drug | 1-Thioglycerol | Millipore Sigma | Cat #: M6145 | |
| Chemical compound, drug | 2-Mercaptoethnol; 2-ME | Thermo Fisher Scientific | Gibco; Cat #: 21985023 | |
| Chemical compound, drug | L-Ascorbic acid; ascorbate | Millipore Sigma | Cat #: A89 60 | |
| Chemical compound, drug | L-Proline; proline | Millipore Sigma | Cat #: P5607 | |
| Chemical compound, drug | ML329 | Cayman Chemical | Cat #: 2248 | |
| Chemical compound, drug | Dexamethasone; Dex | Millipore Sigma | Cat #: D4902 | |
| Chemical compound, drug | GSK1016790A; GSK101 | Sigma-Aldrich | Cat #: G0798 | |
| Chemical compound, drug | GSK205 | AOBIOUS | Cat #: AOB1612 1263130-79-5 | |
| Chemical compound, drug | Sulfinpyrazone | Sigma-Aldrich | Cat #: S9509-5G | |
| Chemical compound, drug | 1,9-Dimethylmethylene blue; DMMB | Sigma-Aldrich | Cat #: 341088 | |
| Software, algorithm | pClamp software suite | Molecular Devices | RRID:SCR_011323 | |
| Software, algorithm | Fiji software – ImageJ | This paper | RRID:SCR_002285; version 2.1.0 | Used to analyze fluorescence confocal imaging of calcium signaling |
| Software, algorithm | MATLAB – Hertz model | *Darling et al., 2006* | | Used to analyze AFM data to determine modulus |
| Software, algorithm | bcl2fastq | Ilumina | RRID:SCR_015058 | |
| Software, algorithm | Ensembl release 76 primary assembly with STAR | *Dobin et al., 2013* | RRID:SCR_002344; version 2.5.1a | |
| Software, algorithm | Subread:featureCount | *Liao et al., 2014* | RRID:SCR_012919; version 1.4.6-p5 | |
| Software, algorithm | Salmon | *Patro et al., 2017* | RRID:SCR_017036; version 0.8.2 | |
| Software, algorithm | RSeQC | *Wang et al., 2012* | RRID:SCR_005275; version 2.6.2 | |
| Software, algorithm | DESeq2 R package | *Love et al., 2014* | RRID:SCR_015687 | |
| Software, algorithm | Pheatmap R package | *Kolde, 2015* | RRID:SCR_016418 | |
| Software, algorithm | ggplot2 R package | *Wickham, 2009* | RRID:SCR_014601 | |
| Software, algorithm | GraphPad Prism, version 9.1 | GraphPad Software, Boston, MA | RRID:SCR_002798; version 9.1.0 | |

*Appendix 1 Continued on next page*

*Appendix 1 Continued*

| Reagent type (species) or resource | Designation | Source or reference | Identifiers | Additional information |
|---|---|---|---|---|
| Software, algorithm | VennDiagram R package | *Chen and Boutros, 2011* | RRID:SCR_002414 | |
| Software, algorithm | g:profiler | *Raudvere et al., 2019* | RRID:SCR_006809 | |
| Software, algorithm | tidyverse R package | *Altman and Krzywinski, 2017* | RRID:SCR_019186 | |
| Software, algorithm | Cytoscape String | *Doncheva et al., 2019*; *Shannon et al., 2003* | RRID:SCR_003032 | |
| Other | Essential 8 Flex Media; E8 | Thermo Fisher Scientific | Gibco; Cat #: A2858501 | hiPSC medium (see Materials and methods: hiPSC culture) |
| Other | Iscove's Modified Dulbecco's Medium, glutaMAX; IMDM | Thermo Fisher Scientific | Gibco; Cat #: 31980097 | Mesodermal differentiation medium (see Materials and methods: Mesodermal differentiation) |
| Other | Ham's F-12 nutrient mix, glutaMAX; F12 | Thermo Fisher Scientific | Gibco; Cat #: 31765092 | Mesodermal differentiation medium (see Materials and methods: Mesodermal differentiation) |
| Other | Penicillin–streptomycin; P/S | Thermo Fisher Scientific | Gibco; Cat #: 15140122 | Mesodermal and chondrogenic differentiation medium supplement (see Materials and methods: Mesodermal differentiation, Chondrogenic differentiation with 3D pellet culture) |
| Other | Insulin–Transferrin–Selenium; ITS+ | Thermo Fisher Scientific | Gibco; Cat #: 41400045 | Mesodermal and chondrogenic differentiation medium supplement (see Materials and methods: Mesodermal differentiation, Chondrogenic differentiation with 3D pellet culture) |
| Other | Chemically defined concentrated lipids | Thermo Fisher Scientific | Cat #: 11905031 | Mesodermal differentiation medium supplement (see Materials and methods: Mesodermal differentiation) |
| Other | Dulbecco's Modified Eagle Medium/F12, glutaMAX; DMEM/F12 | Thermo Fisher Scientific | Cat #: 10565042 | Chondrogenic differentiation medium (see Materials and methods: Chondrogenic differentiation with 3D pellet culture) |
| Other | Modified Eagle Medium (MEM) with nonessential amino acids; NEAA | Thermo Fisher Scientific | Gibco; Cat #: 11140050 | Chondrogenic differentiation medium supplement (see Materials and methods: Chondrogenic differentiation with 3D pellet culture) |
| Other | Fetal bovine serum; FBS | Atlanta Biologicals | Cat #: S11550 | Neutralization medium (see Materials and methods: Chondrogenic differentiation with 3D pellet culture) |
| Other | Axopatch 1D patch-clamp amplifier and digitized with Digidata 1320 digitizer | Molecular Devices | | Patch clamping equipment (see Materials and methods: Patch clamping) |
| Other | Soda lime glass | Kimble Chase | Cat #: 2502 | Patch clamping equipment (see Materials and methods: Patch clamping) |
| Other | Sutter P-86 puller | Sutter Instruments | | Patch clamping equipment (see Materials and methods: Patch clamping) |
| Other | HEPES | Thermo Fisher Scientific | Gibco; Cat #: 15630130 | Calcium signaling medium (see Materials and methods: TRPV4 agonists and antagonists, Patch clamping) |
| Other | Confocal microscope | Zeiss | LSM 880 | Calcium signaling equipment (see Materials and methods: Confocal imaging of $Ca^{2+}$ signaling) |
| Other | Optimal cutting temperature; OCT | Sakura Finetek | Cat #: 4583 | AFM materials (see Materials and methods: AFM measurement of neocartilage mechanical properties) |
| Other | Cryofilm | Section-Lab | Type: 2C(10) | AFM materials (see Materials and methods: AFM measurement of neocartilage mechanical properties) |

*Appendix 1 Continued*

| Reagent type (species) or resource | Designation | Source or reference | Identifiers | Additional information |
|---|---|---|---|---|
| Other | Atomic force microscopy; AFM | Asylum Research | Cat #: MFP-3D Bio | AFM equipment (see Materials and methods: AFM measurement of neocartilage mechanical properties) |
| Other | Silicon cantilever with a spherical tip | Novascan Technologies | | 5 µm diameter, $k \sim 7.83$ N/m; AFM materials (see Materials and methods: AFM measurement of neocartilage mechanical properties) |
| Other | RIPA buffer | Cell Signaling Technology | Cat #: 9806S | Western blot materials (see Materials andmethods: Western blot) |
| Other | Protease inhibitor | Thermo Fisher Scientific | Cat #: 87786 | Western blot materials (see Materials and methods: Western blot) |
| Other | TidyBlot Western Blot Detection Reagent:HRP; TidyBlot-Reagent-HRP | Bio-Rad | Cat #: STAR209 | 1:1000; Western blot materials (see Materials and methods: Western blot) |
| Other | 10% sodium dodecyl sulfate–polyacrylamide gel electrophoresis gel with pre-stained molecular weight markers | Bio-Rad | Cat #: 161-0374 | Western blot materials (see Materials and methods: Western blot) |
| Other | iBright FL1000 Imaging System | Thermo Fisher Scientific | | Western blot equipment (see Materials and methods: Western blot) |
| Other | DNase | Norgen Biotek | Cat #: 25720 | RNA sequencing materials (see Materials and methods: Genome-wide mRNA sequencing) |
| Other | RNA Clean-Up and Concentration Kit | Norgen Biotek | Cat #: 43200 | RNA sequencing materials (see Materials and methods: Genome-wide mRNA sequencing) |
| Other | NovaSeq 6000 | Illumina | | RNA sequencing equipment (see Materials and methods: Genome-wide mRNA sequencing) |
| Other | Safranin-O solution; Saf-O | Millipore Sigma | Cat #: HT904 | Histology materials (see Materials and methods: Histology) |
| Other | Harris hematoxylin with glacial acetic acid; hematoxylin | Poly Scientific | Cat #: 212A16OZ | Histology materials (see Materials and methods: Histology) |
| Other | Vector hematoxilyn QS counterstain | Vector Laboratories | Cat #: H-3404 | Histology materials (see Materials and methods: Histology) |

