## [Editor Report]

Analysis of different types of TRPV4 mutant hiPS cells (mild V620I vs severe T89I mutations) showed alterations in calcium channel function and chondrocyte differentiation. hiPSC-derived chondrocytes with the V620I mutation exhibited increased basal currents through TRPV4, while both mutations showed more rapid calcium signaling with a reduced overall magnitude in response to TRPV4 agonist GSK1016790A compared to wild-type cells. These findings provide potential therapeutic targets for developing treatments for TRPV4-mediated skeletal dysplasias.

---

## [Decision Letter]

**Decision letter after peer review:**

Thank you for submitting your article "Skeletal dysplasia-causing TRPV4 mutations suppress the hypertrophic differentiation of human iPSC-derived chondrocytes" for consideration by *eLife*. Your article has been reviewed by 3 peer reviewers, including Di Chen as the Reviewing Editor and Reviewer #3, and the evaluation has been overseen by a Reviewing Editor and Mone Zaidi as the Senior Editor.

Essential revisions:

1) The authors need to consider analyzing changes in TRPV4 phosphorylation in V620I and T89I mutant cells.

2) In order to confirm the changes in chondrocyte hypertrophy, the authors need to analyze the expression of other markers, such as Runx2, MMP13, and Ihh.

3) To confirm RNA-seq data, RT-PCR verification is recommended.

4) Additional data are required to support the conclusion that the mutations of V620I and T89I are indeed different. Only minor differences were identified from the current study.

*Reviewer #1 (Recommendations for the authors):*

1. In addition to evaluating the electrophysiology of chondrocytes as a functional assessment, the analysis of differences in TRPV4 activation i.e. phosphorylation (like the study by Cao et al. in JBC 2018; 294:5307) should be considered to confirm V620I and T89I mutation-induced changes in TRPV4 activities.

2. The results of 42-day-induction chondrocytes were either little- or non-described in this report. It seems like the data when included make it more challenging to interpret the findings. Will the authors consider repeating the experiment to verify the results or simply removing the data?

3. The key finding shown here is the TRPV4 mutations inhibit chondrocyte hypertrophy. In addition to cell size and COL10 expression, it should be considered to include other data such as the expression of markers, RUX2, MMP13, and IHH for the confirmation of chondrocyte hypertrophy.

4. It is known that RNA-seq measures a rough expression level of a transcript so RT-PCR verification is often performed. Therefore, the RT-PCR analysis of expression levels of the identified genes, ALPL, AMELX, IFITM5, and others, should be considered as an addition to Figure 6.

*Reviewer #2 (Recommendations for the authors):*

In this manuscript, Dicks et al. generated two human iPSC lines with TRPV4 mutations (mild V620I or lethal T89I) using a CRISPR-Cas9 approach and examined their channel function and differentiation abilities into chondrocytes. While their initial goal is to elucidate the detailed molecular mechanisms underlying how these two mutations lead to strikingly distinct severities of skeletal dysplasias, most of their data found that these two mutations behave in a similar manner. The minor differences they found are: 1) increased basal currents in V620I cells; 2) reduced mechanical properties of cartilage matrix in V620I chondrocytes; 3) some differences in DEGs of RNA-seq data. They also stated that "The severe T89I mutation inhibits chondrocyte hypertrophy more than moderate V620I 298 mutation" (page 16). However, no substantiated data were provided to support this conclusion. While a serial of RNA-seq experiments were performed to explore the underlying mechanism, they were not followed by validation experiments to pinpoint the exact pathways or molecular mechanisms. Thus, although using CRISPR-Cas9 and iPSCs are novel and potentially important, this manuscript is overall descriptive with limited mechanistic information.

To the editor, I am not familiar with channel function assays. Please find an expert to review this part.

*Reviewer #3 (Recommendations for the authors):*

TRPV4 is an ion channel protein and mutations in TRPV4 gene resulted in different types of skeletal defects. In this study, the authors created two types of TRPV4 mutations (mild V620I and lethal T89I mutations) in human iPS cells through CRISPR-Cas9 gene editing. They found that mutations of TRPV4 accelerated calcium signaling and reduced response of calcium signaling to TRPV4 agonist. In addition, the V620I mutation led to decreased mechanical properties of cartilage matrix later in chondrogenesis. TRPV4 mutations also upregulated HOX genes and downregulated antioxidant genes, such as CAT and GSTA1, through entire chondrogenesis process. BMP4-induced chondrocyte hypertrophy was also inhibited in TRPV4 mutant cells. This study provided novel information about the functions of TRPV4 in chondrogenesis and help us understand the skeletal dysplasia observed in patients with TRPV4 mutations.

The study was well designed with novel approach and state-of-the -art technology. The experiments were well executed and information gained from this study is valuable. To further improve this manuscript, few suggestions need to be considered.

1)In this study, several chndrogenic marker genes were altered in TRPV4 mutant cells and these genes are regulated by Runx2, such as Col10a1. It is not clear if Runx2 expression is changed in TRPV4 mutant cells.

2)The authors found changes in Wnt3a, Wnt7a and Wnt7b expression in TRPV4 mutant cells. Do the authors know if β-catenin protein levels were altered in TRPV4 mutant cells.

---

## [Author Response]

Essential revisions:1) The authors need to consider analyzing changes in TRPV4 phosphorylation in V620I and T89I mutant cells.

We appreciate the suggestion to look at alterations in TRPV4 phosphorylation in the mutated V620I and T89I lines and how that may be affecting TRPV4 activation. Since we know phosphorylation of TRPV4 by PKC and PKA results in altered sensitization (Cao et al., 2018; Fan, Zhang, & McNaughton, 2009; Peng et al., 2010), we investigated their gene expression using our day-28 RNAseq dataset. *PRKCA*, the gene encoding for protein kinase C α, was upregulated in V620I compared to WT at day 28. Therefore, phosphorylation of TRPV4 by PKC could explain the increased basal calcium signaling in V620I. Unfortunately, there are no commercially available antibodies to perform a western blot and investigate phosphorylated TRPV4; however, we believe this could make for an interesting follow-up study. This data has been added to Figure 4 – Figure S1B, and its relationship to TRPV4 signaling is discussed.

2) In order to confirm the changes in chondrocyte hypertrophy, the authors need to analyze the expression of other markers, such as Runx2, MMP13, and Ihh.

We agree with the reviewers’ suggestions, and we have included these additional data. We believe that showing gene expression and protein production of RUNX2, *MMP3*, IHH, and ALPL will further support the hypertrophic differences in WT, V620I, and T89I. We have shown the gene expression using the RNA-seq data and further confirmed translation and protein production with western blots.

BMP4 treatment significantly increased expression of *COL10A1*, *IHH*, and *ALPL* in all three lines compared to their TGFβ3 controls; however, it only increased expression of *RUNX2* in WT. Surprisingly, *MMP13* expression was significantly decreased by BMP4 treatment in V620I and T89I lines compared to TGFβ3 controls, but there was no change in WT. WT treated with BMP4 had significantly higher expression of *COL10A1*, *RUNX2*, *IHH*, *MMP13*, and *ALPL* compared to V620I and T89I treated with BMP4. The moderate mutation, V620I, treated with BMP4 also had significantly higher expression of *IHH* and *ALPL* compared to the severe mutation, T89I.

These data were consistent with the western blot results. There was an increase of COL10A1, ALPL, IHH, RUNX2, and RUNX2-9 proteins with BMP4 treatment in all three lines. However, BMP4 treatment had a much stronger effect on increasing expression in WT. In support of our hypothesis that T89I had a stronger inhibitory effect on chondrogenic hypertrophy, as this mutation had lower protein production of COL10A1, ALPL, and IHH in response to BMP4 treatment compared to V620I, also consistent with the RNA-seq data. We saw the same trend in MMP13 protein and gene expression, with BMP4 causing a decrease in expression in the mutants but no change in the WT.

To strengthen our argument, we have added the western blot data to Figure 5 and included the RNA seq analysis, full blots, and quantification of the blots to Figure 5 – Figure S1.

3) To confirm RNA-seq data, RT-PCR verification is recommended.

We thank the reviewers for the suggestion to validate the RNA-seq data with RT-qPCR. Several groups have published that RNA-seq methods are valid and more quantitative than RT-qPCR methods, and thus do not benefit from RAT-qPCR verification (Coenye, 2021; Everaert et al., 2017). However, to confirm this point, we performed RT-qPCR on chondrogenic genes during chondrogenesis in addition to the RNA-seq performed on day-28 pellets. Here, we show that when the RT-qPCR data is analyzed in reference to either undifferentiated hiPSCs or to day-28 WT, we see similar expression patterns as the RNA-seq between WT, V620I, and T89I. Additionally, western blots were performed on hypertrophic proteins (e.g., COL10A1, ALPL, IHH, MMP13, RUNX2), and the data were consistent with the BMP4 RNA-seq data set (Figure 5 – Figure S1). Therefore, we believe that the gene expression we report based on our RNA-seq results for both day-28 and day-56 chondrogenic pellets, including those treated with BMP4, are valid.

**Author response image 1. sa2fig1:** 

4) Additional data are required to support the conclusion that the mutations of V620I and T89I are indeed different. Only minor differences were identified from the current study.

We appreciate the reviewer’s suggestion to dive deeper into the differences between V620I- and T89I-mutated chondrocytes. We investigated top differentially expressed genes, compared to WT, that were unique to either V620I or T89I. We found that the two lines became more divergent with continued differentiation to day 56. Upregulated genes unique to V620I included interferon induced protein with tetratricopeptide repeats 3 (*IFIT3*), interferon-induced GTP-binding protein Mx1 (*MX1*), p53 upregulated regulator of p53 levels (*PURPL*), and protein kinase C α (PKC; *PRKCA*) at day 28 and 56. Downregulated genes were related to DNA- and RNA-binding such as zinc finger proteins (*ZNF736, ZNF717,* and *ZNF594*) and ribosomal protein S4 y-linked 1 (*RPS4Y1*) at day 28 and 56. T89I had uniquely upregulated genes such as micro–RNA *MIR1245A*, dickkopf WNT signaling pathway inhibitor 2 (*DKK2*), and carbonic anhydrase 2 (*CA2*) at day 28 and insulin growth factor-like family member 3 (*IGFFL3)*, matrix extracellular phosphoglycoprotein (*MEPE*), annexin A8 (*ANXA8*), *S100A3*, and β catenin (*CTNNB1*) at day 56. Downregulated genes included bone sialoprotein II (*IBSP*) and limb development transcription factor *SP9* at day 28 and various zinc finger proteins at both day 28 and 56.

We also saw differences between the two lines after BMP4 treatment. This was highlighted by the unique association with biological processes for T89I compared to WT and V620I. We propose that V620I slows hypertrophic differentiation, consistent with the moderate phenotype, while T89I inhibits it, consistent with the severe phenotype. This was shown with the protein production of hypertrophic markers ALPL and IHH. While both mutants have lower production compared to WT, T89I has less than V620I.

Reviewer #1 (Recommendations for the authors):1. In addition to evaluating the electrophysiology of chondrocytes as a functional assessment, the analysis of differences in TRPV4 activation i.e. phosphorylation (like the study by Cao et al. in JBC 2018; 294:5307) should be considered to confirm V620I and T89I mutation-induced changes in TRPV4 activities.

We appreciate the suggestion to look at alterations in TRPV4 phosphorylation in the mutated V620I and T89I lines and how that may be affecting TRPV4 activation. Since we know phosphorylation of TRPV4 by PKC and PKA results in altered sensitization (Cao et al., 2018; Fan, Zhang, & McNaughton, 2009; Peng et al., 2010), we investigated their gene expression using our day-28 RNAseq dataset. *PRKCA*, the gene encoding for protein kinase C α, was upregulated in V620I compared to WT at day 28. Therefore, phosphorylation of TRPV4 by PKC could explain the increased basal calcium signaling in V620I. Unfortunately, there are no commercially available antibodies to perform a western blot and investigate phosphorylated TRPV4; however, we believe this could make for an interesting follow-up study. This data has been added to Figure 4 – Figure S1B, and its relationship to TRPV4 signaling is discussed.

2. The results of 42-day-induction chondrocytes were either little- or non-described in this report. It seems like the data when included make it more challenging to interpret the findings. Will the authors consider repeating the experiment to verify the results or simply removing the data?

We thank the reviewer for pointing out the confusion with the day-42 data. To improve the clarity of the manuscript, we have moved the data to the supplement. The text and figures in the manuscript have been updated to reflect these changes.

We performed the differentiation through day 42 3-4 times per cell line. From our previous studies, we have found that 28 days is sufficient for chondrogenic differentiation, but we used day 42 to investigate if changes were occurring with prolonged differentiation. This data indicated there were some changes, but we ultimately decided to take the differentiation to day 56, as reported with the RNA-seq data, with and without BMP4 treatment.

3. The key finding shown here is the TRPV4 mutations inhibit chondrocyte hypertrophy. In addition to cell size and COL10 expression, it should be considered to include other data such as the expression of markers, RUX2, MMP13, and IHH for the confirmation of chondrocyte hypertrophy.

We agree with the reviewer that assessing additional hypertrophy markers will further confirm the differences in BMP4-induced hypertrophy. Therefore, we performed a western blot and found consistent results. BMP4 treatment increased protein production of hypertrophy markers ALPL, COL10A1, IHH, and RUNX2’s isoform 9. The changes in protein production with BMP4 were more prominent in WT compared to the mutants. BMP4 treatment also had a stronger effect on ALPL, as well as COL10A1 and IHH, in V620I compared to T89I. These data have been added to Figure 5 and Figure 5 —figure supplement 1, and the text has been updated.

4. It is known that RNA-seq measures a rough expression level of a transcript so RT-PCR verification is often performed. Therefore, the RT-PCR analysis of expression levels of the identified genes, ALPL, AMELX, IFITM5, and others, should be considered as an addition to Figure 6.

We appreciate the reviewer’s suggestion to verify the RNA-seq data with RT-qPCR. Several groups have published that RNA-seq methods are more quantitative than RT-qPCR and do not require RT-qPCR verification (Coenye, 2021; Everaert et al., 2017). Nonetheless, we compared chondrogenic gene expression at day 28 of chondrogenesis and saw similar patterns between RT-qPCR and RNA-seq. The BMP4 RNA-seq data set was further confirmed as we compared gene expression to protein expression with western blots of hypertrophy markers, including ALPL, and saw consistent results (Figure 5 – Figure S1). These data have been added to the manuscript, and we believe the RNA-seq data has been validated.

Reviewer #2 (Recommendations for the authors):In this manuscript, Dicks et al. generated two human iPSC lines with TRPV4 mutations (mild V620I or lethal T89I) using a CRISPR-Cas9 approach and examined their channel function and differentiation abilities into chondrocytes. While their initial goal is to elucidate the detailed molecular mechanisms underlying how these two mutations lead to strikingly distinct severities of skeletal dysplasias, most of their data found that these two mutations behave in a similar manner. The minor differences they found are: 1) increased basal currents in V620I cells; 2) reduced mechanical properties of cartilage matrix in V620I chondrocytes; 3) some differences in DEGs of RNA-seq data. They also stated that "The severe T89I mutation inhibits chondrocyte hypertrophy more than moderate V620I 298 mutation" (page 16). However, no substantiated data were provided to support this conclusion. While a serial of RNA-seq experiments were performed to explore the underlying mechanism, they were not followed by validation experiments to pinpoint the exact pathways or molecular mechanisms. Thus, although using CRISPR-Cas9 and iPSCs are novel and potentially important, this manuscript is overall descriptive with limited mechanistic information.

We thank the reviewer for the summary of the paper. We have further investigated the differences between WT and the two mutant lines to add to the RNA-seq experiments. As suggested by another reviewer, we looked at protein kinase gene expression, which may be altering TRPV4 phosphorylation and ultimately changes in channel activation. This expression data is consistent with the basal calcium differences we saw, and we believe these warrant further investigation in a follow-up study regarding biochemical changes to the channel structure and activation.

We also further validated the differences in BMP4-induced hypertrophy by looking at protein production. BMP4 not only increased hypertrophic proteins COL10A1, ALPL, IHH, and RUNX2, but we saw much larger increases in WT compared to mutants. Further, ALPL production was increased in the moderate V620I mutation compared to the severe T89I mutation, indicating a potential player in the differences in disease severity caused by the two mutants.

Finally, we investigated the DEGs between V620I and T89I to highlight the differences between the two mutations. We believe this study has served as a foundation for identifying potential mechanisms leading to the disease phenotypes of moderate and severe skeletal dysplasias. In future studies, we hope to validate these mechanisms

Reviewer #3 (Recommendations for the authors):TRPV4 is an ion channel protein and mutations in TRPV4 gene resulted in different types of skeletal defects. In this study, the authors created two types of TRPV4 mutations (mild V620I and lethal T89I mutations) in human iPS cells through CRISPR-Cas9 gene editing. They found that mutations of TRPV4 accelerated calcium signaling and reduced response of calcium signaling to TRPV4 agonist. In addition, the V620I mutation led to decreased mechanical properties of cartilage matrix later in chondrogenesis. TRPV4 mutations also upregulated HOX genes and downregulated antioxidant genes, such as CAT and GSTA1, through entire chondrogenesis process. BMP4-induced chondrocyte hypertrophy was also inhibited in TRPV4 mutant cells. This study provided novel information about the functions of TRPV4 in chondrogenesis and help us understand the skeletal dysplasia observed in patients with TRPV4 mutations.The study was well designed with novel approach and state-of-the -art technology. The experiments were well executed and information gained from this study is valuable. To further improve this manuscript, few suggestions need to be considered.1)In this study, several chndrogenic marker genes were altered in TRPV4 mutant cells and these genes are regulated by Runx2, such as Col10a1. It is not clear if Runx2 expression is changed in TRPV4 mutant cells.

We thank the reviewer for their suggestion to investigate RUNX2 expression. We found that *RUNX2* gene expression was significantly increased with BMP4 treatment in the WT line but not in the mutant lines. Additionally, *RUNX2* gene expression was significantly higher in BMP4-treated WT compared to BMP4-treated mutants. We also looked at RUNX2 protein production using western blot, in addition to other hypertrophic markers. We found that not only was RUNX2 expressed in the samples, specifically so was the RUNX2 isoform 9. BMP4-treated WT had higher protein production of both RUNX2 and RUNX2-9 compared to BMP4-treated mutants. BMP4 treatment increased RUNX2-9 production more (than RUNX2) in the mutants compared to their TGFβ3 controls.

2)The authors found changes in Wnt3a, Wnt7a and Wnt7b expression in TRPV4 mutant cells. Do the authors know if β-catenin protein levels were altered in TRPV4 mutant cells.

We appreciate the reviewer’s suggestion to look at β catenin given the differences in *WNT3A, WNT7A*, and *WNT7B* gene expression. We looked at our RNA seq data to identify if there were differences in β-catenin. We found that expression of *CTNNB1* was similar between the three lines at both day 28 and day 56. However, T89I, the severe TRPV4 mutation, had significantly higher *CTNNB1* expression compared to WT after 56 days of chondrogenic differentiation.

References

Adkar, S. S., Wu, C. L., Willard, V. P., Dicks, A., Ettyreddy, A., Steward, N.,... Guilak, F. (2019). Step-Wise Chondrogenesis of Human Induced Pluripotent Stem Cells and Purification Via a Reporter Allele Generated by CRISPR-Cas9 Genome Editing. *Stem Cells, 37*(1), 65-76. doi:10.1002/stem.2931

Camacho, N., Krakow, D., Johnykutty, S., Katzman, P. J., Pepkowitz, S., Vriens, J.,... Cohn, D. H. (2010). Dominant TRPV4 mutations in nonlethal and lethal metatropic dysplasia. *Am J Med Genet A, 152A*(5), 1169-1177. doi:10.1002/ajmg.a.33392

Cao, S., Anishkin, A., Zinkevich, N. S., Nishijima, Y., Korishettar, A., Wang, Z.,... Zhang, D. X. (2018). Transient receptor potential vanilloid 4 (TRPV4) activation by arachidonic acid requires protein kinase A–mediated phosphorylation. *Journal of Biological Chemistry, 293*(14), 5307-5322. doi:10.1074/jbc.m117.811075

Coenye, T. (2021). Do results obtained with RNA-sequencing require independent verification? *Biofilm, 3*, 100043. doi:10.1016/j.bioflm.2021.100043

Dicks, A., Wu, C. L., Steward, N., Adkar, S. S., Gersbach, C. A., & Guilak, F. (2020). Prospective isolation of chondroprogenitors from human iPSCs based on cell surface markers identified using a CRISPR-Cas9-generated reporter. *Stem Cell Research & Therapy, 11*(1), 66. doi:10.1186/s13287-020-01597-8

Everaert, C., Luypaert, M., Maag, J. L. V., Cheng, Q. X., Dinger, M. E., Hellemans, J., & Mestdagh, P. (2017). Benchmarking of RNA-sequencing analysis workflows using whole-transcriptome RT-qPCR expression data. *Scientific Reports, 7*(1). doi:10.1038/s41598-017-01617-3

Fan, H. C., Zhang, X., & McNaughton, P. A. (2009). Activation of the TRPV4 ion channel is enhanced by phosphorylation. *Journal of Biological Chemistry, 284*(41), 27884-27891. doi:10.1074/jbc.M109.028803

Kang, S. S., Shin, S. H., Auh, C.-K., & Chun, J. (2012). Human skeletal dysplasia caused by a constitutive activated transient receptor potential vanilloid 4 (TRPV4) cation channel mutation. *Experimental and Molecular Medicine, 44*(12), 707. doi:10.3858/emm.2012.44.12.080

Peng, H., Lewandrowski, U., Muller, B., Sickmann, A., Walz, G., & Wegierski, T. (2010). Identification of a Protein Kinase C-dependent phosphorylation site involved in sensitization of TRPV4 channel. *Biochem Biophys Res Commun, 391*(4), 1721-1725. doi:10.1016/j.bbrc.2009.12.140

Rock, M. J., Prenen, J., Funari, V. A., Funari, T. L., Merriman, B., Nelson, S. F.,... Cohn, D. H. (2008). Gain-of-function mutations in TRPV4 cause autosomal dominant brachyolmia. *Nature Genetics, 40*(8), 999-1003. doi:10.1038/ng.166

Sun, S. (2012). The Mutation of Transient Receptor Potential Vanilloid 4 (TRPV4) Cation Channel in Human Diseases. In Kang (Ed.), *Mutagenesis*. InTech.

Wu, C. L., Dicks, A., Steward, N., Tang, R., Katz, D. B., Choi, Y. R., & Guilak, F. (2021). Single cell transcriptomic analysis of human pluripotent stem cell chondrogenesis. *Nat Commun, 12*(1), 362. doi:10.1038/s41467-020-20598-y